# GRAM: SPATIAL GENERAL-PURPOSE AUDIO REPRESENTATION MODELS FOR REAL-WORLD APPLICATIONS

## ABSTRACT

Although audio foundation models have seen great progress on a wide variety of tasks, their application in real-world acoustic environments with reverberation and noise has been less successful. Moreover, as audio foundation models are typically trained on dry, single-channel audio clips, the inherent spatial nature of real-world sound scenes is overlooked and tasks involving sound localization are ruled out. To address these limitations, we propose GRAM: a General-purpose Real-world Audio Model utilizing a multi-channel masked auto-encoder approach to efficiently learn spatial audio representations from high-quality simulated real-world scenes. To evaluate the performance of GRAM and other audio foundation models in real-world sound scenes, we release Nat-HEAR: A naturalistic version of the HEAR benchmark suite comprising a simulated real-world version, as well as two new sound localization and RT60 estimation tasks. We show that the performance of GRAM surpasses all state-of-the-art self-supervised audio foundation models and speech models on both HEAR and Nat-HEAR, while using only a fraction of the training data. GRAM also showcases state-of-the-art localization performance, surpassing even supervised sound localization approaches, and can be flexibly applied either to a two-channel, binaural sound format or a four-channel, Ambisonics format. Validating GRAM's performance on real-world sound recordings demonstrates robust transfer to real-world scenes. Taken together, GRAM presents a significant advancement towards robust, spatial audio foundation models for real-world applications. [1]

## 1 INTRODUCTION

Despite the complexity and diversity of everyday sound scenes, human listeners effortlessly interact with their acoustic environment in myriad ways. Audio foundation models that perform a similar, human-like range of tasks have received widespread attention (Turian et al., 2022; Wang et al., 2022a; Yang et al., 2021). While these models demonstrate strong performance on audio benchmarks with minimal fine-tuning (for example, (Chen et al., 2023; Baevski et al., 2020; Yadav et al., 2024), they overlook inherent aspects of real-world sound scenes: the spatial dimension, reverberation and background noise. Specifically, audio foundation models are typically trained on large-scale sound datasets consisting of dry, non-spatial sound clips such as AudioSet (Gemmeke et al., 2017) and Librispeech (Panayotov et al., 2015). The effectiveness of these approaches in naturalistic, complex acoustic environments with background noise and reverberation is therefore limited.

Crucially, the lack of spatial information in audio embeddings precludes sound localization tasks and the use of spatial sound features for improving performance on complex audio tasks such as audio scene analysis. Audio scene analysis refers to the separation of overlapping sound waves in complex multi-source sound scenes and the subsequent grouping of the frequency components into coherent auditory objects (Bregman, 1984; Bizley & Cohen, 2013). In humans, such audio scene analysis is aided by spatial cues as well (Bizley & Cohen, 2013; van der Heijden et al., 2019). Similarly, incorporating spatial knowledge into universal audio embedding models is expected to benefit downstream tasks where ambient intelligence and acoustic awareness are desired, such as acoustic scene understanding.

---

[1]All code and materials are available on `TBD`

A major challenge for the development of audio foundation models for real-world sound scenes is the scarcity of naturalistic sound data for model training. Recording a vast amount of spatial sound scenes with varying reverberation characteristics is infeasible and requires fine-tuned recording setups and conditions (Zheng et al., 2024). The simulation of spatial acoustic scenes has therefore received much attention and progressed from simulating shoebox rooms (Scheibler et al., 2018) to simulating realistic sound scenes from everyday life (Chen et al., 2020). Yet, no large-scale datasets of naturalistic sound scenes exist to date, hampering both the development as well as the systematic evaluation of audio foundation models for real-world sound scenes. For example, benchmark task suites such as HEAR (Turian et al., 2022), HARES (Wang et al., 2022a) and SUPERB (Yang et al., 2021) solely contain datasets consisting of dry, non-spatial sound clips without background noise and do not include spatial reasoning tasks such as sound localization.

To address these limitations of audio foundation models for real-world applications, we present GRAM (General-purpose, Real-world Audio Model). GRAM is a self-supervised, multi-channel masked auto-encoder model that efficiently learns spatial general-purpose audio representations from simulated real-world sound scenes. To train GRAM, we developed a custom pipeline which makes use of the Soundspace 2.0 platform (Chen et al., 2022a) to simulate high-quality real-world sound scenes from AudioSet (Gemmeke et al., 2017), and of WHAMR! (Maciejewski et al., 2020) for adding background noise. Further, to promote the systematic evaluation of audio foundation models on naturalistic sound scenes, we introduce Nat-HEAR. Nat-HEAR is an extension of the HEAR benchmark suite which contains simulated real-world versions of the downstream tasks, and additionally includes two sound localization tasks and two RT60 estimation tasks. We experiment with two state-of-the-art encoder architectures (Transformer and Mamba) to assess which architecture is most suitable for spatial general-purpose audio representation learning in our multi-channel masked auto-encoder approach. We present two versions of GRAM to ensure flexible application across audio formats: GRAM-Binaural for two-channel audio clips, and GRAM-Ambisonics for four-channel audio clips in the first-order Ambisonics format. Finally, we perform systematic ablation experiments on mask type (patch versus time-based), ratio of simulated real-world sound scenes and conventional dry sound clips in pre-training, mask ratio, and in-batch sampling to elucidate which factors are critical for successful spatial general-purpose audio representation learning.

Empirical results demonstrate that GRAM efficiently learns robust and generalizable spatial general-purpose audio representations, outperforming all state-of-the-art audio foundation models and speech models on HEAR and Nat-HEAR. GRAM excels especially at complex tasks such as audio scene analysis and exhibits excellent sound localization performance, outperforming even supervised models trained with auxiliary spatial features. Finally, GRAM demonstrates robust transfer to recordings of real-world sound scenes, overcoming the need for extensive domain-specific adaptations. Our key contributions can be summarized as:

**General-Purpose Audio Foundation Model (GRAM):** We present GRAM, a multi-channel masked auto-encoder that shows state-of-the-art performance on a human-like range of tasks in naturalistic sound scenes, including sound localization.GRAM is the first audio foundation model that is available both for binaural, two-channel audio formats and for four-channel, Ambisonics audio formats.

**A large-scale dataset for high-quality simulations of real-world sound scenes**: We release the full set of binaural room impulse responses (BRIRs) and ambisonics room impulse responses (ARIRs) corresponding to 85,000 naturalistic sound scenes that we used for our naturalistic training pipeline.

**Nat-HEAR:** To encourage systematic evaluation of audio foundation models on naturalistic scenes, we present an extended version of the HEAR benchmark suite (Turian et al., 2022) in which we transform the HEAR datasets in the HEAR downstream tasks to naturalistic versions. Additionally, we add two novel, naturalistic sound localization tasks in Nat-HEAR.

## 2 RELATED WORK

**Supervised audio representation learning:** Supervised methods for audio representation learning have achieved notable success in recent years. Approaches such as AST (Gong et al., 2021a), PaSST (Koutini et al., 2022) and HTS-AT (Chen et al., 2022b) have Transformer-based architectures as a backbone, for example ViT (Dosovitskiy et al., 2021) and Swin Transformer (Liu et al., 2021). To mitigate the need for large annotated datasets, some of these approaches are based on models pre-

trained on image data (for example, AST (Gong et al., 2021a), PSLA (Gong et al., 2021b)). Question and answer models constitute a more recent category of supervised approaches that integrate audio representation learning with large language models (for example, Spatial-AST (Zheng et al., 2024) and Qwen-Audio (Chu et al., 2023). However, supervised training requires large-scale annotated datasets and is sub-optimal for learning general-purpose audio representations that generalize across tasks.

**Self-supervised audio representation learning:** Self-supervised audio representation learning approaches aim to learn robust audio representations that generalize to a wide variety of tasks (Wang et al., 2022a; Turian et al., 2022). Masking-based approaches utilizing transformer backbones to reconstruct masked patches of input spectrograms currently constitute predominant approach, including (SSAST (Gong et al., 2022)), MSM-MAE (Niizumi et al., 2022), MaskSpec (Chong et al., 2023), MAE-AST (Baade et al., 2022) and Audio-MAE (Huang et al., 2022). Of the masked auto-encoder approaches, MW-MAE (Yadav et al., 2024) achieves state-of-the-art performance on the HEAR benchmark by using multi-window local-global attention in the decoder. Recently, SSAM (Yadav & Tan, 2024) utilized a Mamba (Gu & Dao, 2023) architecture in their encoder and achieved similar performance as MW-MAE. In contrast to the masked auto-encoders, BEATS (Chen et al., 2023) utilizes masking-based approach based on latent embeddings extracted by an acoustic tokenizer. Finally, successful self-supervised approaches that do not rely on masking at all include contrastive learning frameworks such as COLA (Saeed et al., 2021).

Another category of self-supervised audio representation models focuses on speech representations specifically, making use of generative, predictive or contrastive learning (Mohamed et al., 2022). These speech models are typically trained on datasets such as Librispeech (Panayotov et al., 2015) or LibriLight (Kahn et al., 2020) and include state-of-the-art models such as Wav2Vec2 (Baevski et al., 2020), HuBERT (Hsu et al., 2021) and WavLM (Chen et al., 2021). However, while these models excel at speech-based tasks, they do not necessarily generalize well to non-speech sounds and non-speech tasks (Turian et al., 2022). Crucially, none of the existing self-supervised approaches for audio or speech representation learning optimize for performance in real-world sound scenes that are spatial, reverberant, and noisy.

# 3 Materials and Methods

## 3.1 Simulating real-world acoustic scenes

**Pipeline overview:** A room impulse response (RIR) captures room specific acoustic properties such as reverberation. We utilized high-resolution, detailed 3D meshes of houses with various architectural characteristics from Matterport3D (Chang et al., 2017) in order to simulate RIRs for many different rooms in each house with the Monte Carlo ray tracing RIR simulator provided by SoundSpaces 2.0 Chen et al. (2022a). SoundSpaces 2.0 combines the simulated RIRs with a head-related transfer function (HRTF) (Algazi et al., 2001) to generate a binaural RIR (BRIR) or with an ambisonics microphone configuration to generate an ambisonics RIR (ARIR). BRIRs capture both room acoustic properties and human spatial hearing characteristics introduced by the shape of the ears, head and torso, while ARIRs capture room acoustic properties as well as the spatial cues encoded in first-order Ambisonics.

**Components of simulated real-world scenes:** Matterport3D contains scans of 90 houses. We discarded five houses for which meshes were not of sufficient quality. For each of the remaining 85 houses, we simulated 1,000 real-world scenes. Each scene consisted of a randomly sampled listener location (microphone location for ambisonics), sound source location and noise source location in the room. For BRIRs (binaural), we randomly sampled head orientation from a range $[0°, 360°]$. We placed the sound source location at a randomly sampled location with respect to the listener or microphone (distance range $[1.5 \text{ m}, 5 \text{ m}]$; azimuth range $[0°, 360°]$; elevation range $[-90°, +90°]$). Noise was either localized (50% of the scenes) or diffuse (50% of the scenes). For localized noise, we randomly sampled one location in the room. For diffuse noise, we randomly sampled three, four or five locations in the room. We then rendered a set of RIRs to describe all components in the naturalistic scene. Given sound source location $s$, listener (microphone) location $r$, and receiver head orientation $\theta$, we rendered RIRs describing the sound path from the source to the listener (microphone) as $\text{BRIR}(s, r, \theta)$ and as $\text{ARIR}(s, r, \theta)$. Given a number of noise sources $n_i$ at noise source location $\phi_i$, listener location $r$, and receiver head orientation $\theta$, we rendered the RIR

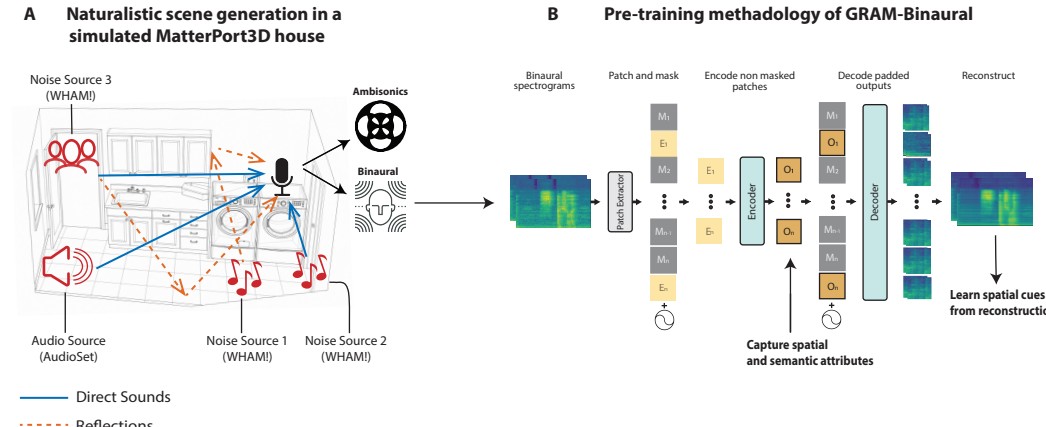

**A** Naturalistic scene generation in a simulated MatterPort3D house

**B** Pre-training methodology of GRAM-Binaural

Figure 1: Proposed self-supervised approach for training GRAMs on naturalistic binaural scenes. (A) We generate binaural and ambisonics naturalistic scenes using SoundSpaces2.0 simulator (Chen et al., 2022a) in MatterPort3D houses. These scenes contain realistic reverberations, and diffused/localized noise interference. (B) The self-supervised approach for learning audio representation with spatial attributes. The Patch Extraction layer patches and embeds the input spectrogram using 2D convolution. A random subset of patches is masked (ratio = 0.8). Unmasked patches are fed to the encoder. The decoder takes the encoder outputs padded with the masked patches and reconstructs the original spectrogram. For the ambisonics spectrograms, the methodology stays the same except that inputs now contains 4 channel mel spectrograms, and intensity vectors (IVs).

describing the path from the noise source(s) to the listener as $BRIR_i(\phi_i, r, \theta)$ and as $ARIR(\phi_i, r, \theta)$. This procedure resulted in a total of 85,000 sets of BRIRs as well as 85,000 sets of ARIRs (see Appendix A for all parameters).

## 3.2 GRAM FRAMEWORK

The GRAM learns spatial audio representation by reconstructing multi-channel masked spectrogram patches. First, a patch extractor consisting of a single convolutional layer with 2D convolutional filters divides each multi-channel spectrogram into $n$ non-overlapping patches $P_1, \ldots, P_n$ with $P_i \in \mathbb{R}^{C \times T \times F}$, and embeds each patch into a linear patch embedding $E_i \in \mathbb{R}^{768}$ (Figure 1). Non-masked patch embeddings are input to the encoder, for which we selected the 12-layer ViT-Base (ViT-B) Transformer (Dosovitskiy et al., 2021) similar to Huang et al. (2022); Yadav et al. (2024). The encoder outputs patch representations $O_i \in \mathbb{R}^{768}$ for $i = 1, \ldots, n$, where $n$ is the number of unmasked patches. Finally, a Transformer decoder with local-global attention (Yadav et al., 2024) followed by a linear head takes all patch representations $O_1, \ldots, O_n$ as well as all masked patches $M_1, \ldots, M_n$ to reconstruct the multi-channel spectrogram from last layer embeddings.

## 3.3 PRE-TRAINING

**Online mixing of naturalistic sound scenes:** The 85,000 naturalistic scenes were split into a train set of 70,000 scenes (corresponding to 70 Matterport3D houses), and a test set of 15,000 scenes (15 Matterport3D houses) for down-stream evaluation (see Section 3.4). We used the 70,000 naturalistic scenes in the train set to generate naturalistic scenes for all audio clips in the unbalanced training set of AudioSet (10-second sound tracks of 1.74 million YouTube videos (Gemmeke et al., 2017)). Specifically, during training we randomly paired an AudioSet clip with a noise sound clip from the WHAMR! background noise database (Maciejewski et al., 2020). WHAMR! noise clips longer than 10 s were trimmed to 10 s duration and a linear fade-in/fade-out of 200 ms was applied to every noise clip prior to mixing of the sound scene.

To create a naturalistic sound scene, we then convolved the AudioSet clip either with $BRIR(s, r, \theta)$ for GRAM-Binaural, or with a $ARIR(s, r, \theta)$ for GRAM-Ambisonics, to obtain $T$. Similarly, we convolved the WHAMR! noise clip with the $BRIR(\phi_i, r, \theta)$ to obtain $N_i$. In naturalistic scenes

with diffuse background noise, the diffuse noise field $D$ was generated by summing the noise clips $D = \sum_{i=1}^{M} N_i$ where $N_i$ are individual noise clips and $M$ is the total number of noise clips. The naturalistic sound scene $S$ was then calculated as $S = T + bN$ for scenes with localized noise and as $S = T + bD$ for scenes with a diffuse noise field. Here, $b$ is a scaling parameter introduced to mix target and noise sound clips at a given signal-to-noise ratio (SNR) ranging between +5 dB and +40 dB.

**Input features:** We transformed the channels of each sound scene (i.e., the waveforms) into log-scale mel spectrograms using 128 mel filters in the frequency range of 50-16000 Hz with a 25 ms Hanning window and 10 ms hop length, resulting in spectrograms of dimension $1001 \times 128$, later we added zero padding to achieve dimension of $1024 \times 128$. For GRAM-Ambisonic, we extracted normalized active Intensity Vectors (IVs) from the spectrograms as additional input features encoding spatial information (see Appendix B). We concatenated mel spectrograms and intensity vectors, resulting in input $x = [x_{mel}, IVs]$ for each naturalistic scene generated from an AudioSet clip.

**In-batch sampling:** As the online mixing of naturalistic acoustic scenes is computationally expensive due to multiple long convolutions, we used a random in-batch sampling procedure to increase the effective batch size in a computationally efficient manner. We randomly sampled 16 partially overlapping segments of 2 seconds to create 16 samples of dimension $200 \times 128$. This increases the original batch size of 96 to an effective batch size of 1536.

**Patch extraction and masking:** For pre-training, we divided the binaural spectrogram into $P_i \in \mathbb{R}^{2 \times 8 \times 16}$, and ambisonics spectrograms into $P_i \in \mathbb{R}^{7 \times 8 \times 16}$ patches. We used an adapted version of the mask-based framework of MW-MAE (Yadav et al., 2024), randomly selecting a subset of $n$ patches $M_1, \ldots, M_n$ for $i = 1, \ldots, n$ for masking (masking ratio = 0.8) and replacing their embedding with a learnable mask token. Finally, we added fixed sinusoidal positional embeddings to all embedded patches.

**Decoder with local-global attention:** The decoder takes as input both the unmasked patches $O_1, \ldots, O_n$ with $O_i \in \mathbb{R}^{768}$, and the masked patches $M_1, \ldots, M_n$ with $M_i \in \mathbb{R}^{768}$ as well as fixed sinusoidal positional embeddings for each patch (Figure 1). To implement local-global attention (Yadav et al., 2024), we selected window sizes of [2, 5, 10, 25, 50, 100, 0, 0]. Here, 0 signifies global plain attention.

**Pre-training specification:** We trained all GRAMs for 500 K steps on an H100 92 GB GPU machine with 16 CPU cores. We used the AdamW optimizer (Loshchilov & Hutter, 2017) with weight decay rate of 0.01, gradient clipping, and a cosine learning rate scheduler with 10 K steps warm-up. The initial learning rate was set to 0.0002, and decayed to 0. We optimize the mean squared error (MSE) loss function between the predicted masked patches and their corresponding input spectrogram patches.

### 3.4 EXPERIMENTS

**Model comparison:** We compare the performance and efficiency of GRAM-Binaural, GRAM-Ambisonics on downstream tasks with state-of-the-art self-supervised audio representation models with a similar number of parameters as GRAM (90 M): MAE-16x16 (Huang et al., 2022), SSAST-patch (Gong et al., 2022), BEATs-iter3 (Chen et al., 2023), MW-MAE-B-200-4x16 (Yadav et al., 2024), SSAM (Yadav & Tan, 2024); self-supervised speech representation models Wav2Vec 2.0 Base (Baevski et al., 2020), HuBERT Base (Hsu et al., 2021), WavLM Base (Chen et al., 2021). To quantify the impact of pre-training with naturalistic sound scenes, we further train GRAM-Clean. GRAM-Clean follows the same experimental setup as the GRAM-Binaural and GRAM-Ambisonics with the distinction of only consuming dry audioset audio clips.

**Downstream tasks:** We evaluate GRAM and other state-of-the-art models on the HEAR benchmark task suite, which presents a wide range of tasks to evaluate the downstream performance of audio representation models (Turian et al., 2022). To avoid redundancy we selected the same subset of HEAR tasks as previously used in (Yadav et al., 2024). To enable in-depth evaluation of audio scene analysis capabilities, we added the time-stamp based sound event detection task DCASE-2016 Task 2 (Mesaros et al., 2018) from the HEAR benchmark suite.

We additionally evaluated performance on simulated real-world sound scenes using Nat-HEAR, which provides a naturalistic version of all selected datasets in the HEAR benchmark suite in

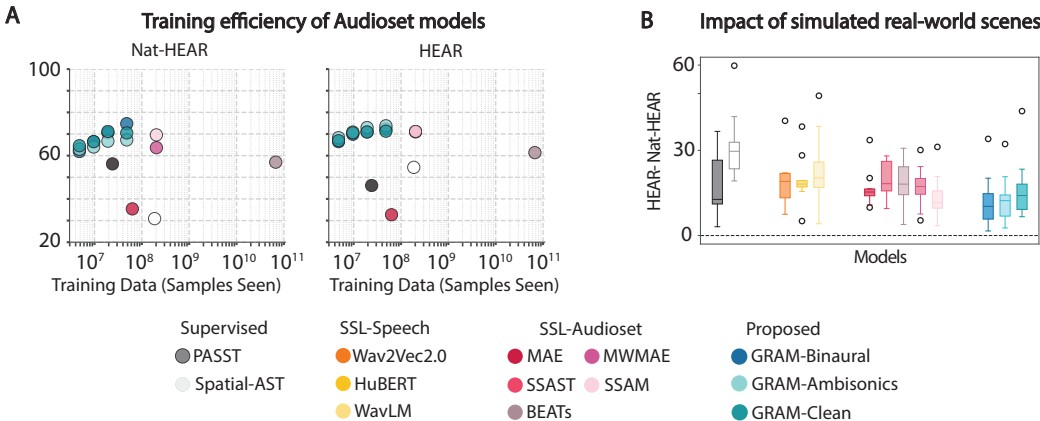

Figure 2: **Downstream model performance on naturalistic sound scenes**. (A) Nat-HEAR and HEAR downstream performance as a function of training data quantity. (B) Box plots of the difference in performance on HEAR and Nat-HEAR, excluding the DCASE-2016 task. Box limits reflect the first and third quartile, center line the median (see also Table 7).

two audio formats: a two-channel, binaural format and a four-channel, first-order Ambisonics format. We included sound localization tasks for two different domains which we generated using HEAR benchmark datasets: A speech localization task based on SC-5, and an environmental sound localization task based on ESC-50. The localization tasks are modeled as a multi-output regression task in which model outputs represent the estimated 3D Cartesian coordinates $[x, y, z]$ on the unit sphere (Adavanne et al., 2018). Finally, to assess the transferability of GRAM to real-world sound scenes, we evaluate also on the sound event detection and localization tasks in TUT Sound Events 2018 REAL (Adavanne et al., 2019), and STARSS23 (Shimada et al., 2023).

**Downstream evaluation:** To evaluate the single-channel SOTA audio representation models on Nat-HEAR, we utilized the omnidirectional channel $W$ of the first-order Ambisonics (Zotter & Frank, 2019) version of Nat-HEAR as model input. The outputs of MAE, SSAST and BEATs were not suitable for the time-stamp based DCASE-2016 sound event detection task. Hence, these models were not evaluated on the DCASE-2016 task. Further, we included Spatial-AST (Zheng et al., 2024) as this is the only model trained on spatial sound scenes and therefore the sole model evaluated on the Nat-HEAR localization tasks besides GRAM-Binaural and GRAM-Ambisonics. To evaluate two-channel models such as GRAM-Binaural on HEAR, we duplicated the single-channel spectrograms of the original HEAR to generate model compatible input. Following the HEAR protocol (Turian et al., 2022) for downstream task evaluation, we extracted embeddings from the frozen pretrained models and trained a shallow downstream classifier on these embeddings to assess how well the learned representations generalize to a broad range of tasks. The embedding extraction for GRAM is described in Appendix E.

**Quantifying overall performance:** We calculate for each model $m$, the score $s(m)$ to give an impression of the overall performance, similar to (Yang et al., 2021). This score reflects a model's improvement with respect to the maximum improvement over the baseline obtained by the current state-of-the-art model, averaged across all tasks included in the benchmark. This metric effectively ranks the improvement of models over the baseline as a function of the current maximum improvement (see Appendix E). We use the HEAR-Naive baseline based on mel-spectrograms (Turian et al., 2022). Furthermore, we calculated the average score over all tasks.

**Evaluating sound localization performance:** We evaluate the sound localization performance on the newly generated sound localization tasks in Nat-HEAR by calculating the Direction of Arrival (DoA) error $\theta$ between the $[x, y, z]$ coordinates of the target sound source on the unit sphere using the arc cosine of the dot product of the unit vectors: $\theta = \arccos(v \cdot \hat{v})$. Note that we included a third GRAM framework for the localization tasks besides GRAM-Binaural and GRAM-Ambisonics, which is GRAM-Binaural with time-based masking instead of patch-based masking. In particular, as explained in Section 3.3 we carry out ablations on masking type. Here, for localization, we hypothesized that time-based masking may lead to better localization results for GRAM-Binaural

than patch-based masking as it enables the model to learn representations along the entire frequency axis, similar to human spatial hearing (van der Heijden et al., 2019; Carlile et al., 1999).

## 3.5 Generalization to real world scenes

To evaluate GRAM-Ambisonics on real world scenes, we conduct experiments on two datasets: TUT Sound Events 2018 (Adavanne et al., 2019) and STARSS23 (Shimada et al., 2023). TUT Sound Events 2018 consists of simulated spatial audio generated by convolving dry audio clips with measured RIRs. In contrast, STARSS23 contains real-world ambisonics recordings captured with spatial microphone arrays, enabling assessment of model transferability to in-the-wild conditions. Notably, STARSS23 features polyphonic scenes with moving sources and environmental noise, presenting a substantially more challenging downstream task.

**TUT Sound Events 2018:** We resample all audio to 32kHz and extract segments with corresponding localization annotations. We formulate localization as a polar coordinate regression task, predicting azimuth and elevation angles $[\theta, \phi] \in [0, 360) \times [-90, 90]$. We follow the HEAR protocol to evaluate our representations.

**STARSS23:** We utilized the audio-only subset of STARSS23, as our model does not incorporate visual modalities. Following standard preprocessing, we resample waveforms to 32kHz and adopt the Activity-Coupled Cartesian Direction of Arrival (ACCDOA) framework (Shimada et al., 2021). This framework jointly models sound event detection and localization across 13 sound classes. A class is considered active at frame $t$ when the predicted Cartesian coordinate magnitude $\|\mathbf{c}_t\| > 0.5$, where $\mathbf{c}_t \in \mathbb{R}^3$ represents the unit direction vector. We extracted frame-level embeddings from GRAM-Ambisonics (Appendix E) yielding representations at 80ms intervals. To match STARSS23's 100ms label resolution, we apply adaptive 1D temporal pooling over embeddings. A linear projection head then maps pooled representations to per-frame predictions for both sound event detection (SED) and direction of arrival (DOA) estimation.

**Training Protocols:** To assess the pre-trained capabilities of GRAM-Ambisonics, we evaluated three training regimes: (1) full fine-tuning of all model parameters, (2) linear probing with frozen encoder weights, and (3) training from scratch. All protocols shared the batch size 512, and 100 training epochs. All other experimental settings follow the SELD baseline model (Shimada et al., 2023). For training from scratch protocol, we assesed four learning rates [1e-3, 1e-4, 2e-4, 5e-4], other protocols had a learning rate of 1e-4.

## 3.6 Ablations

To establish crucial factors for successful spatial general-purpose audio representations learning, we carried out a series of ablation experiments. For GRAM-Binaural, we trained also an encoder with a state-of-the-art 8-layer Mamba architecture (Gu & Dao, 2023; Yadav & Tan, 2024) to assess the optimal architecture choice for spatial general-purpose audio representation learning. To ensure that computational overhead and model capacity were comparable between the Transformer and Mamba encoder, we used similar parameter counts. We also tested the impact of mask type for GRAM-Binaural, comparing patch-based masking as described above to time-based masking. For time-based masking, patches were defined as $P_i \in \mathbb{R}^{2 \times 2 \times 128}$ such that they spanned the entire frequency range. For time-based masking, we used window sizes and [2, 5, 10, 25, 50, 0, 0, 0] to implement local-global attention in the decoder. For both GRAM-Binaural and GRAM-Ambisonics, we assessed the optimal ratio ($\lambda$) between simulated real-world sound scenes and clean, dry sound clips in pretraining data for $\lambda = 0.0, 0.25, 0.5, 0.75, 1.0$. Finally, we examined various masking ratios [0.4, 0.6, 0.8, 0.9] and in-batch sampling factors [4, 8, 16] for GRAM-Binaural. For all ablations, GRAM-Binaural and - if applicable - GRAM-Ambisonics were trained with the exact same parameters specified above, except the masking ratio ablation, where we reduced the effective batch size from 1536 to 384.

## 4 Results

**Performance on simulated real-world sound scenes (Nat-HEAR):** Table 1 demonstrates that GRAM-Binaural ($s = 74.8$, Avg. = 73.9) and GRAM-Ambisonics ($s = 70.5$, Avg. = 71.1) learn robust

Table 1: Performance on Nat-HEAR. Reported values reflect the average performance $\pm$ standard deviation, calculated using $n$-fold cross-validation as specified by the HEAR. Bold numbers indicate the best performing model on the specific task. SSAST* is trained on both AudioSet and Librispeech. Tasks are specified in Appendix C.

| Model | Acoustic Events and Scene Analysis | | | | Speech | | | Music | | | | s(m) | Avg. |
|---|---|---|---|---|---|---|---|---|---|---|---|---|---|
| | DCASE | FSD50K | LC | ESC-50 | CD | VL | SC-5 | NS | BO | Mri-S | Mri-T | | |
| **Baseline** | | | | | | | | | | | | | |
| HEAR-Naive | 26.5 | 8.7 | 27.4±1.6 | 17.2±2.2 | 32.3±2.2 | 11.7±2.2 | 12.0 | 75.6 | 84.3±4.5 | 68.6±1.3 | 60.5±1.3 | 0.0 | 38.6 |
| **Speech SSL** | | | | | | | | | | | | | |
| Wav2Vec2 | 32.0 | 23.0 | 54.6±1.9 | 36.4±2.9 | 48.6±0.6 | 27.2±1.6 | 78.9 | 15.2 | 71.2±6.4 | 75.7±0.5 | 45.9±0.6 | 32.5 | 46.2 |
| HuBERT | 57.6 | 26.6 | 52.5±2.2 | 49.5±2.2 | 57.4±1.1 | 46.8±3.4 | 89.2 | 16.0 | 77.1±6.0 | 78.2±0.7 | 52.4±1.6 | 45.2 | 54.8 |
| WavLM | 25.3 | 20.5 | 52.1±0.6 | 41.4±2.1 | 52.3±1.5 | **47.9±4.6** | 89.9 | 11.2 | 61.4±7.2 | 69.3±0.9 | 39.0±2.0 | 37.8 | 46.4 |
| **AudioSet SSL** | | | | | | | | | | | | | |
| MAE | – | 27.9 | 53.2±1.0 | 65.7±1.2 | 48.5±1.3 | 19.0±1.5 | 57.4 | 53.4 | 79.2±7.8 | 81.0±4.9 | 56.5±12.3 | 34.5 | 54.2 |
| SSAST* | – | 15.6 | 41.6±2.4 | 44.8±1.0 | 39.7±2.9 | 12.7±1.3 | 19.9 | 52.0 | 81.8±3.6 | 76.5±3.6 | 64.6±1.5 | 17.5 | 44.9 |
| BEATs | – | 46.5 | 63.7±1.2 | 72.6±3.9 | 54.8±1.6 | 27.5±4.3 | 83.5 | 54.2 | 70.3±6.2 | 83.2±1.0 | 71.0±1.4 | 55.7 | 62.7 |
| MW-MAE | 83.8 | 44.3 | 64.8±1.1 | 69.7±5.6 | 59.3±1.0 | 31.8±1.8 | 86.7 | 59.2 | 77.1±3.6 | 90.1±0.8 | 73.9±0.6 | 62.5 | 67.3 |
| SSAM | 70.0 | 46.0 | 63.2±1.1 | 73.1±2.4 | 62.3±1.0 | 38.8±2.6 | 86.2 | 65.4 | 84.3±7.0 | **92.6±0.4** | 76.8±1.0 | 68.4 | 68.9 |
| GRAM-Binaural | **93.0** | 52.8 | **72.3±0.7** | 82.6±3.2 | **63.3±1.3** | 35.1±3.8 | **91.0** | **67.6** | 85.6±5.1 | 91.7±0.9 | **78.3±1.3** | **74.8** | **73.9** |
| GRAM-Ambisonics | 90.2 | 49.5 | 68.8±0.9 | 79.4±2.7 | 61.4±0.9 | 36.4±4.2 | 87.2 | 64.6 | 83.4±4.7 | 91.3±0.6 | 78.1±1.4 | 70.5 | 71.8 |
| GRAM-Clean | 90.9 | 50.5 | 66.4±0.8 | 80.0±2.4 | 62.0±1.3 | 32.2±2.3 | 87.3 | 65.2 | 82.2±5.6 | 90.2±0.8 | 75.1±0.7 | 67.3 | 71.1 |
| **Supervised** | | | | | | | | | | | | | |
| PASST | – | **56.9** | 52.1±1.9 | **89.7±2.1** | 49.9±1.0 | 18.4±2.3 | 61.1 | 16.0 | **93.6±4.0** | 85.5±1.7 | 55.6±3.0 | 56.2 | 57.9 |
| Spatial-AST | – | 40.0 | 49.9±1.5 | 70.1±3.3 | 41.6±0.5 | 11.7±2.7 | 54.8 | 50.2 | 77.1±2.8 | 77.7±0.9 | 55.0±1.6 | 30.9 | 52.8 |

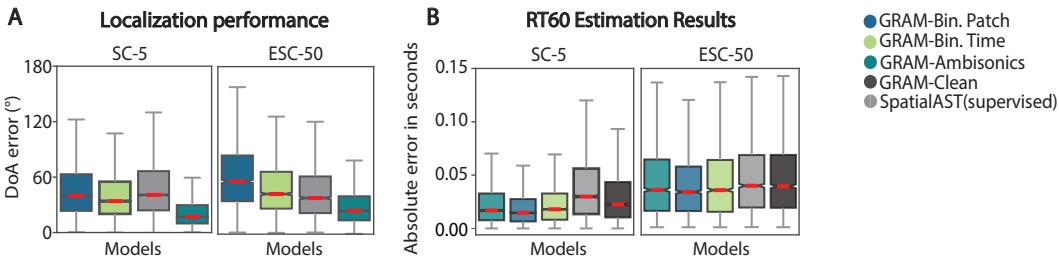

Figure 3: Localizing sound sources in simulated real-world sound scenes, and estimating RT60s. (A) Boxplots of direction of arrival (DoA) error. (B) Boxplots of RT60 estimation error (absolute error). All box limits: first and third quartile; center line: median; whiskers: 1.5 times the interquartile range.

general-purpose audio representations, outperforming all other self-supervised audio representation models on Nat-HEAR. Moreover, GRAMs requires substantially less training data to achieve state-of-the-art performance (Figure 2).

Comparing the performance on Nat-HEAR to the performance on HEAR (that is, clean and dry sounds) highlights the degradation in performance that audio representation models experience in naturalistic sound scenes (Figure 2A). However, GRAM-Binaural and GRAM-Ambisonics exhibit a relatively small drop in performance, highlighting that the model performs the tasks in simulated real-world sound scenes almost as well as the same tasks on clean sounds. Further, the success of our naturalistic training pipeline is highlighted by the difference in degradation between the multi-channel GRAMs (GRAM-Binaural and GRAM-Ambisonics) and GRAM-Clean: GRAM-Binaural and GRAM-Ambisonics drop less in performance than GRAM-Clean (Figure 2B).

**Performance on dry, non-spatial and clean sound scenes (HEAR):** We find that all GRAMs surpasses all other self-supervised audio representation models on HEAR (Table 7). GRAM-Clean achieved state-of-the art performance ($s = 73.8$, Avg. = 83.1), followed by GRAM-Binaural ($s = 72.3$, Avg. = 82.5) and GRAM-Ambisonics ($s = 71.3$, Avg. = 81.1). The superior performance on HEAR of GRAM-Binaural and GRAM-Ambisonics over other audio representation models indicates that training on simulated real-world scenes does not reduce downstream task performance on clean, dry sound scenes.

**Sound localization and RT60 estimation in simulated real-world sound scenes:** GRAMs exhibit excellent localization capabilities in simulated real-world sound scenes, despite the presence of

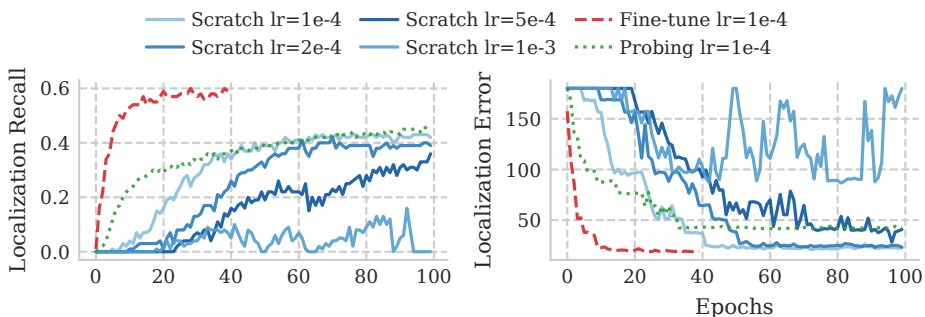

Figure 4: **Fine-tuning dynamics on STARSS23.** Validation metrics across training epochs for three protocols: full fine-tuning, linear probing, and training from scratch. The naturalistic pre-training helps GRAM-Ambisonics train faster and better. Using different learning rates, or increasing training epochs does not improve the GRAM-Ambisonics-scratch performance.

Table 2: TUT Sound Events 2018 Real dataset (DoA error). SeldNET (Adavanne et al., 2019), PILOT (Schymura et al., 2021), Spatial Libri (Sarabia et al., 2023), ELSA (Devnani et al., 2024)

| MODEL | REAL ↓ |
|---|---|
| **Supervised** | |
| SeldNET | $26.6°$ |
| PILOT | $4.2°$ |
| **Self-Supervised** | |
| Spatial Libri. | $12.4°$ |
| ELSA | $15.0°$ |
| GRAM-Amb. | $11.3°$ |

Table 3: Comparison of SELD scores for STARSS23 dataset. Native sampling rate is 24kHz, whereas upsampled is 32kHz. SOTA (Wang et al., 2023), Baseline (Shimada et al., 2023)

| MODEL | $ER_{20°} ↓$ | $F_{20°} ↑$ | $LE_{CD} ↓$ | $LR_{CD} ↑$ |
|---|---|---|---|---|
| **Native sampling rate** | | | | |
| SOTA | 0.42 | 59.0 % | $13.7°$ | 72.0% |
| Baseline | 0.57 | 29.9% | $22.0°$ | 47.7% |
| **Upsampled** | | | | |
| GRAM-Amb. (Fine-tune) | 0.51 | 41.4% | $18.6°$ | 60.5% |
| Baseline (Reprod.) | 0.62 | 28.3% | $23.7°$ | 45.7% |

reverberation and background noise in the scenes (Figure 3). We find that time-based masking is indeed more successful for sound localization with GRAM-Binaural than patch-based masking. Crucially, the localization acuity of GRAM-Ambisonics is substantially higher than that of Spatial-AST for both the speech dataset (SC-5) and the sound scene dataset (ESC-50), even though Spatial-AST is a supervised model trained with location labels. Furthermore, GRAMs with spatial attributes estimated RT60s statistically significantly better to GRAM-Clean and Spatial-AST.

**Generalization to localization in recorded real-world sound scenes:** Table 2 shows that GRAM-Ambisonics generalizes successfully to the real-world sound scenes in the TUT Sound Events 2018 dataset (Adavanne et al., 2019), obtaining a lower DoA error than other self-supervised models. GRAM-Ambisonics even outperforms supervised models such as SeldNET (Adavanne et al., 2018) (note that PILOT (Schymura et al., 2021) is sound localization model trained directly on TUT Sound Events 2018). Furthermore, GRAM-Ambisonics demonstrate the transferability of our pre-trained weights for on STARSS23 dataset. Specifically, comparing the validation curves of pre-trained GRAMs in Figure 4 reveal that pre-trained weights carry substantial information regarding sound event detection and sound localization on challenging environments. Furthermore, Table 3 reveals that GRAM-Ambisonics achieves competetive results on STARSS23 dataset even with non-native sampling rate, additional data augmentations, and domain specific architecture.

### 4.1 ABLATION STUDIES

**Ratio of clean and naturalistic data in pretraining:** Prior work on learning spatially aware audio representations from spectrograms demonstrated that pretraining on a mixture of clean and naturalistic sound scenes rather than on naturalistic sound scenes only, benefits the quality of learned representation (Devnani et al., 2024). We therefore investigated to what extent pretraining on a mixture of clean and naturalistic sound scenes affected the performance of GRAM-Binaural and

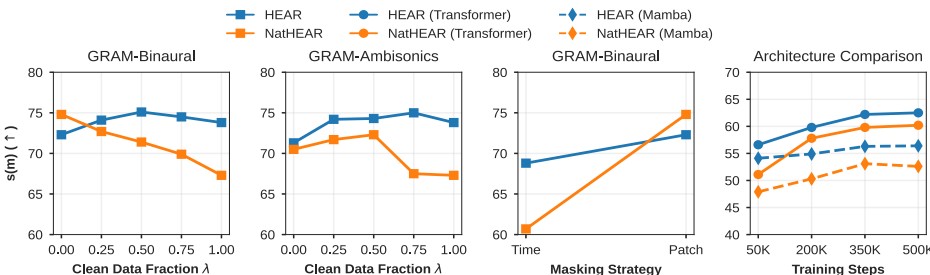

Figure 5: **Ablation Studies**. Effect of hyperparameters on HEAR and Nat-HEAR Performance. From left to right; (1) GRAM-Binaural downstream performance as a function of the ratio $\lambda$ between clean and naturalistic scenes in the pre-training data. (2) GRAM-Ambisonics downstream performance as a function of the ratio $\lambda$ between clean and naturalistic scenes in the pre-training data. (3) effect of masking strategy to downstream performance for GRAM-Binaural (4) comparison of Mamba and Transformer architectures on binaural training data. Important to note that architectures depicted in (4) was trained on reduced batch size ($96 \rightarrow 32$).

GRAM-Ambisonics on HEAR and Nat-HEAR. The panels on the left in Figure 5 show that the performance of GRAM-Binaural on Nat-HEAR increases with lower $\lambda$, while performance on HEAR is optimal with a mixture of clean and naturalistic scenes ($\lambda = 0.5$). GRAM-Ambisonics performed best on Nat-HEAR using a mixture of clean and naturalistic scenes ($\lambda = 0.5$) in line with Devnani et al. (2024). In contrast, GRAM-Ambisonics performed better on HEAR with more clean data during pretraining (high $\lambda$).

**Masking strategy:** Figure 5 illustrates that patch-based masking results in better downstream performance on both HEAR and Nat-HEAR. However, as shown in Figure 4, time-based masking leads to more accurate localization for GRAM-Binaural and may therefore still be considered as masking strategy depending on the purpose of the model.

**Encoder architecture:** As shown in Figure 5, an encoder with a Transformer backbone consistently performed better than an encoder with a Mamba backbone both on clean downstream tasks (HEAR) and on naturalistic downstream tasks (Nat-HEAR).

## 5 DISCUSSION AND CONCLUSION

We present a General-purpose, Real-world Audio representation Model (GRAM), which learns spatial audio representations using a multi-channel masked auto-encoder approach. GRAM demonstrates remarkable performance in naturalistic sound scenes as well as clean sound scenes, surpassing all state-of-the-art self-supervised spectrogram-based audio foundation models while requiring only a fraction of the training data. Moreover, GRAM is the first audio foundation model that is available in both a two-channel, binaural format and a four-channel, first-order ambisonics format. GRAM successfully encoded spatial information into the learned audio representations, outperforming both self-supervised and supervised approaches on sound localization and RT60 estimation tasks. We furthermore release Nat-HEAR: a naturalistic version of the HEAR benchmark suite including also localization tasks. In sum, GRAM is a new state-of-the-art audio representation model that incorporates high-quality spatial learning and exhibits robust performance in real-world sound scenes, representing a crucial step towards successful real-world applications of audio foundation models.

**Limitations and future work**: Although GRAM performs well on HEAR speech tasks in comparison to other self-supervised models trained on AudioSet, we plan to train GRAM on a mixture of speech and general audio data (e.g., AudioSet) in order to assess GRAMs capability for speech learning in more detail. Furthermore, GRAM opens a path towards multi-modal spatial learning and can serve as a basis for downstream applications such as audio-visual scene representation learning (Mahmud & Marculescu, 2023), robotics Ledder et al. (2025), and audio-language representation learning (Zheng et al., 2024; Chu et al., 2023).

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

APPENDIX

## A  SOUNDSPACES 2.0 SPECIFICATIONS

We generate our BRIRs using the simulator provided by SoundSpaces 2.0 (Chen et al., 2022a). Our hyperparameters for the simulator is depicted in Table 4.

Table 4: Acoustic configuration parameters utilized in SoundSpaces 2.0 to generate our BRIRs.

| Parameter | Value | Parameter | Value |
|---|---|---|---|
| directSHOrder | 3 | indirectSHOrder | 3 |
| sampleRate | 32000 | frequencyBands | 8 |
| maxDiffractionOrder | 10 | transmission | True |
| indirect | True | indirectRayCount | 15000 |
| indirectRayDepth | 400 | sourceRayCount | 200 |
| sourceRayDepth | 20 | threadCount | 16 |
| agentHeigth | 1.5m | | |

## B  INTENSITY VECTORS

Following Devnani et al. (2024); Wang et al. (2022b), we extracted intensity vectors utilizing the equation below:

$$\boldsymbol{I}_{\text{active}}(t, f) = \Re \left[ A_{0,0}^*(t, f) \begin{pmatrix} A_{1,\text{-}1}(t, f) \\ A_{1,0}(t, f) \\ A_{1,1}(t, f) \end{pmatrix} \right], \tag{1}$$

where $A_{n,m}$ are the $n^{\text{th}}$ and $m^{\text{th}}$ order and mode of the ambisonics signal corresponding to its omnidirectional ($W$) and three dipole ($Z, Y, X$) components, and $(\cdot)^*$ denotes complex conjugation. IVs are scaled to unit-norm.

## C  HEAR AND NAT-HEAR TASKS

Table 5 illustrates the abbreviations, task description, and the type that we have utilized to benchmark our models.

Table 5: Overview of the HEAR and Nat-HEAR tasks.

| Abbreviation | Task Name | Description | Type |
|---|---|---|---|
| DCASE | DCASE-2016 Task 2 (Mesaros et al., 2018) | Event detection of overlapping office sounds in synthetic mixtures | Scene Analysis |
| FS50K | FSD50k (Fonseca et al., 2022) | Multilabel, large scale audio tagging | Scene Analysis |
| LC | LibriCount (Stöter et al., 2018) | Speaker Count Identification, Simulated Cocktail Party | Scene Analysis |
| ESC-50 | ESC-50 (Piczak) | Environmental Sound Classification | Environmental Sound Classification |
| CD | Crema-D (Cao et al., 2014) | Emotion Recognition | Speech Analysis |
| VL | VoxLingua107 Top10 (Valk & Alumäe, 2021) | Spoken language identification | Speech Analysis |
| SC-5 | Speech Command 5h (Warden, 2018) | Keyword Spotting, reduced training subset | Speech Analysis |
| NS | NSynth Pitch 5h (Engel et al., 2017) | Pitch Classification, reduced training subset | Pitch Classification |
| BO | Beijing Opera (Tian et al., 2014) | Classifying percussion instruments | Percussion |
| Mri-S | Mridangam Stroke (Anantapadmanabhan et al., 2013) | Stroke classification in pitched percussion instruments | Percussion |
| Mri-T | Mridangam Tonic (Anantapadmanabhan et al., 2013) | Tonic classification in pitched percussion instruments | Percussion |

## D  RT60 ESTIMATION TASKS

Nat-HEAR includes two RT60 estimation tasks (ESC-50 and SC-5) in addition to the direction-of-arrival estimation tasks. For synthesizing these tasks, we did not add additional localized/diffused noise. Specifically, we convolved ESC-50 and SC-5 clips with BRIR$(s, r, \theta)$ for Nat-HEAR Binaural, or with ARIR$(s, r, \theta)$ for Nat-HEAR Ambisonics. We estimated the ground truth RT60s using the first channel of the ARIR. To estimate the RT60s, we utilized the Schroeder method (Schroeder, 1965) from Pyroomacoustics package (Scheibler et al., 2018). Explicitly, we measure the RT30 and extrapolate to RT60 using the decay curve.

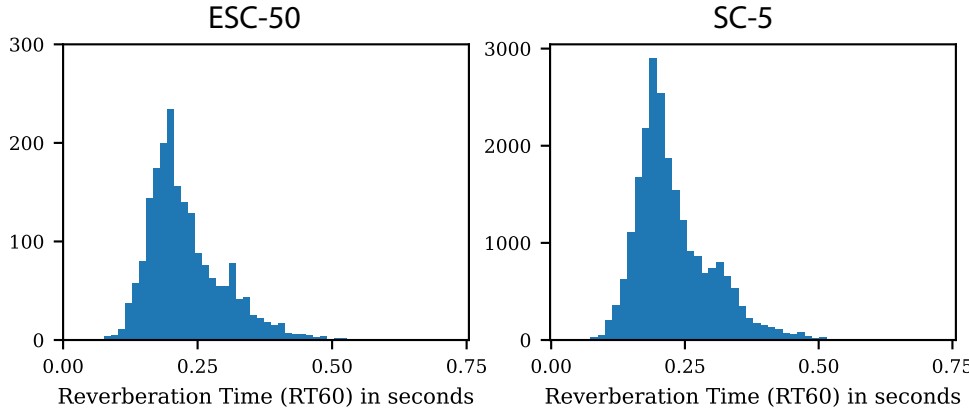

Figure 6: Distribution of the estimated RT60s for ESC-50 and SC-5 datasets.

Figure 6 depicts the RT60 distributions of ESC-50 and SC-5 datasets. Furthermore Table 6 presents the median absolue errors that we got with GRAMs on ESC-50 and SC-5 tasks.

Table 6: Absolute median error comparison on RT60 estimation tasks.

(a) SC-5

| Model | Median Error |
|---|---|
| GRAM-T-Clean | 0.0225 |
| GRAM-T-Ambisonics | 0.0169 |
| GRAM-T-Binaural (Patch) | 0.0146 |
| GRAM-T-Binaural (Time) | 0.0179 |
| Spatial-AST | 0.0299 |

(b) ESC-50

| Model | Median Error |
|---|---|
| GRAM-T-Clean | 0.0461 |
| GRAM-T-Ambisonics | 0.0421 |
| GRAM-T-Binaural (Patch) | 0.0397 |
| GRAM-T-Binaural (Time) | 0.0418 |
| Spatial-AST | 0.0468 |

## E    EXTRACTING GRAM EMBEDDINGS FOR DOWNSTREAM TASKS

We extracted GRAM embeddings for downstream evaluations by encoding embeddings for all patches $P_1, \ldots, P_n$ using the GRAM encoder. We used the exact patch aggregation process as in (Niizumi et al., 2022). Audio clips were split into non-overlapping 2-second chunks and the embedded patches concatenated over time. Later, we took the mean over the time axis to generate scene embeddings independent of the input audio duration. Finally, to evaluate GRAMs on the localization tasks, we used [CLS] embeddings of the 2-second samples, and averaged them to create scene embeddings for localization tasks.

## F    DOWNSTREAM PERFORMANCE METRIC

Similar to the procedure in SUPERB (Yang et al., 2021), let $s_t$ be the metric for task $t$. We then calculate the generalizability metric $\text{HEAR}_s(m)$, and $\text{Nat-HEAR}_s(m)$ for model $m$ as:

$$s(m) = \frac{100}{T} \sum_t^T \frac{s_t(m) - s_t(baseline)}{s_t(SOTA) - s_t(baseline)}$$

Intuitively, this metric ranks the improvement of models over the baseline as a function of the maximum improvement over the baseline obtained by the current state-of-the-art. Note that we replace $s_t(m)$ for task $t$ of model $m$ with 0 when the model scores below baseline performance for task $t$. Similarly, when $s_t(SOTA)$ is lower than baseline for task $t$, we set for all models $s_t$ for this task to 0. In this way, all values are restricted to a range of improvement between 0% and 100%.

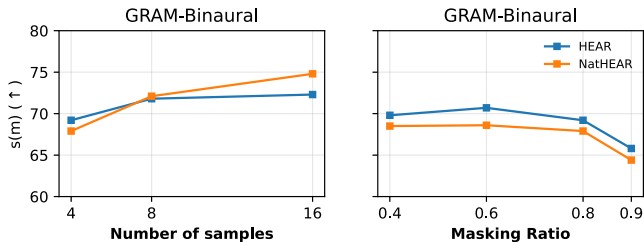

Figure 7: Additional ablation studies. Effect of hyperparameters on HEAR and Nat-HEAR Performance. From left to right; (1) GRAM-Binaural downstream performance as a function of the number of in batch samples. (2) The effect of masking ratio for GRAM-Binaural. Important to note that GRAM-Binaural depicted in (2) was trained on reduced number of samples ($16 \rightarrow 4$).

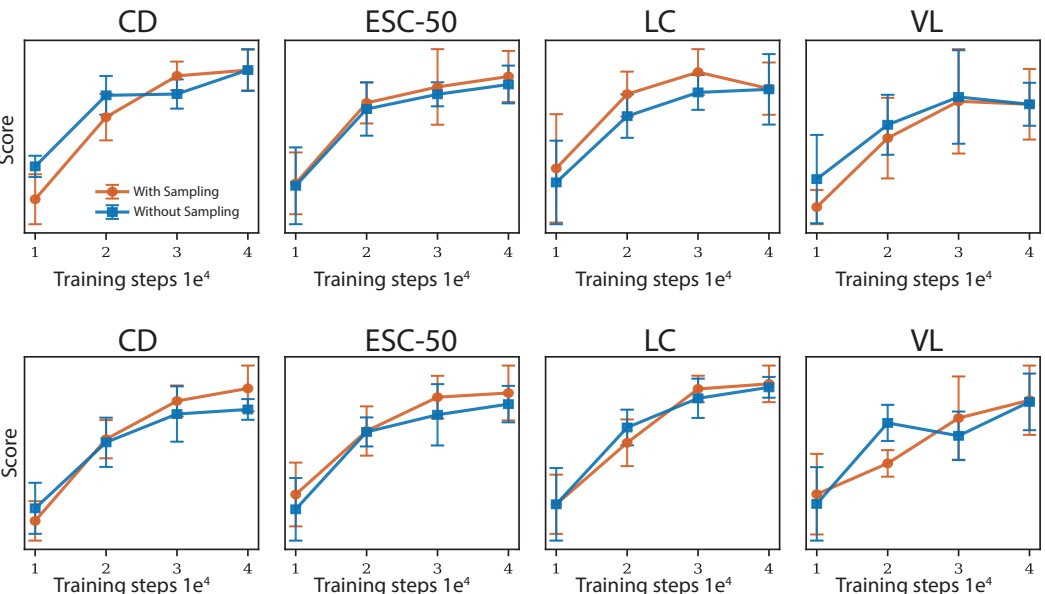

Figure 8: Additional ablation studies. Effect of in batch sampling on HEAR and Nat-HEAR performance when the effective batch size is kept the same. From top to bottom; (1) GRAM-Binaural downstream performance on HEAR as a function of the in-batch sampling (2) GRAM-Binaural downstream performance on Nat-HEAR as a function of the in-batch sampling

## G  ADDITIONAL ABLATION STUDIES

Firstly, we further investigated the masking ratio, and in batch sampling as a function of HEAR and Nat-HEAR performance. Secondly, we investigated the localization performance in terms of mixture of naturalistic and clean audio $\lambda$. Thirdly, we investigated the localization performance in terms of noise levels in the NatHEAR benchmark, which is low [20-40dB], medium [10-20dB] and high [5-10]dB. Lastly, we looked at the effect of in batch sampling when effective batch size is kept constant. For this experiment, we used gradient accumulation over 16 batches. Consequently number of in batch samples were set to 16, yielding effective batch size of 512 for both models.

**In-batch sampling:** Figure 7 1 depicts that in-batch sampling helped immensely with the downstream performance on both HEAR and Nat-HEAR downstream. Increasing the number of in-batch samples leads to higher batch sizes with minimal computational constraints. Furthermore, Figure 8 shows that in-batch sampling does not result in a drop in downstream performance or model convergence.

**Masking ratio:** Figure 7 2 depicts that optimal masking ratio is 0.6 for HEAR and Nat-HEAR performance, and higher masking ratios, such as 0.9 harms the performance.

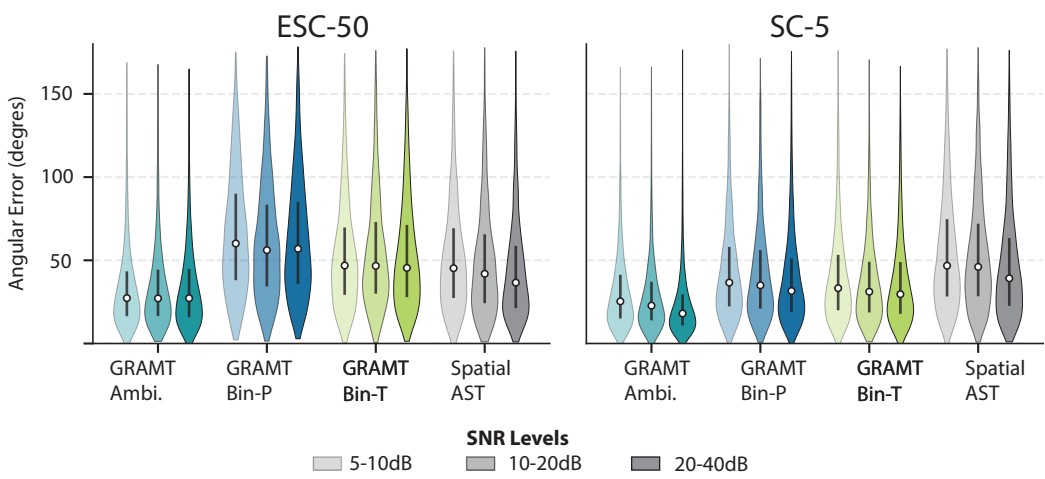

Figure 9: Ablation study on noise levels and localization performance of GRAMs, and Spatial-AST.

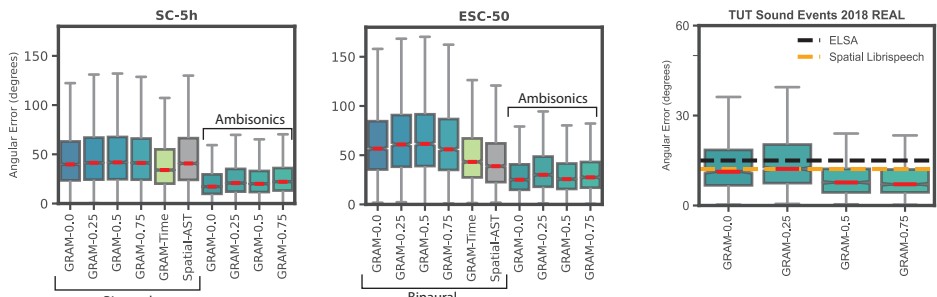

Figure 10: Detalized localization scores. From left to right, Panel 1 demonstrates the localization performance of GRAM-Binaural and GRAM-Ambisonics with tested $\lambda$ parameters on SC-5h. Panel 2 demonstrates the localization performance of GRAM-Binaural and GRAM-Ambisonics with tested $\lambda$ parameters on ESC-50. Lastly, Panel 3 demonstrates the localization performance of GRAM-Ambisonics with tested $\lambda$ parameters on TUT Sound Events Real compared to other self-supervised methods.

**Localization performance:** Figure 10 depicts that GRAM-Ambisonics achieve the highest performance on SC-5h, ESC-50, and TUT Sound Events 2018 REAL compared to other baselines regardless of the fraction of naturalistic scenes. Importantly, we do not observe a correlation between $\lambda$ and localization performance, suggesting that GRAMs learn to exploit spatial attributes with little data.

**Noise Levels:** Figure 9 depicts that GRAM-Ambisonics achieves the highest performance on SC-5h, ESC-50, and TUT Sound Events 2018 REAL compared to other baselines regardless of the fraction of naturalistic scenes. Importantly, we do not observe a correlation between $\lambda$ and localization performance, suggesting that GRAMs learn to exploit spatial attributes with little data.

## H   RESULTS ON ORIGINAL HEAR BENCHMARK SUITE

We evaluated our models on the dry, non anechoic, and non-spatialized HEAR Benchmark suite. Table 7 depicts the achieved results on the HEAR sub tasks.

Table 7: Performance comparison of audio representation models across HEAR tasks. All values represent the HEAR scores with standard deviation where available. Bold numbers indicate the best performing model on the specific task. SSAST* is trained on both AudioSet and Librispeech.

| Model | Acoustic Events and Scene Analysis | | | | Speech | | | Music | | | | | | |
| | DCASE | FSD50K | LC | ESC-50 | CD | VL | SC-5 | NS | BO | Mri-S | Mri-T | s(m) | Avg. |
|---|---|---|---|---|---|---|---|---|---|---|---|---|---|
| **Baseline** | | | | | | | | | | | | | |
| HEAR-Naive | 8.8 | 13.2 | 43.5 ± 1.6 | 28.6 ± 3.1 | 38.0 ± 2.3 | 14.8 ± 3.0 | 13.3 | 87.6 | **98.7** ± 1.9 | 94.1 ± 0.5 | 87.6 ± 6.4 | 0.0 | 48.0 |
| **Speech SSL** | | | | | | | | | | | | | |
| Wav2Vec 2.0 | 23.5 | 29.4 | 69.9 ± 2.1 | 46.4 ± 1.8 | 57.3 ± 1.1 | 34.9 ± 2.4 | 85.3 | 17.4 | 81.4 ± 4.8 | 90.7 ± 0.8 | 77.0 ± 0.9 | 30.7 | 55.7 |
| HuBERT | 78.3 | 32.8 | 63.3 ± 1.2 | 58.6 ± 2.8 | 71.2 ± 1.2 | **65.2** ± 2.9 | **94.0** | 19.8 | 93.2 ± 5.9 | 94.6 ± 0.4 | 85.0 ± 2.5 | 43.6 | 68.7 |
| WavLM | 27.0 | 25.7 | 61.3 ± 2.3 | 49.5 ± 3.8 | 64.3 ± 1.3 | 60.1 ± 3.2 | 93.8 | 18.2 | 84.3 ± 6.3 | 88.8 ± 1.0 | 76.8 ± 0.5 | 36.1 | 59.1 |
| **AudioSet SSL** | | | | | | | | | | | | | |
| MAE | – | 33.4 | 62.3 ± 1.1 | 72.9 ± 2.1 | 60.8 ± 1.8 | 21.3 ± 5.8 | 66.6 | 63.6 | 94.5 ± 5.6 | 94.8 ± 0.6 | 85.1 ± 10.4 | 31.3 | 65.5 |
| SSAST* | – | 21.4 | 57.8 ± 3.3 | 58.3 ± 2.6 | 48.0 ± 2.1 | 15.4 ± 2.6 | 22.0 | 64.2 | 95.8 ± 4.3 | 90.2 ± 5.9 | 89.1 ± 8.0 | 15 | 56.2 |
| BEATs | – | 54.1 | 77.8 ± 1.2 | 85.8 ± 2.9 | 66.9 ± 2.5 | 39.7 ± 4.3 | 86.9 | 68.6 | 94.1 ± 3.5 | 95.5 ± 0.4 | 96.6 ± 0.5 | 59.2 | 76.6 |
| MW-MAE | 94.2 | 51.8 | 80.3 ± 1.9 | 82.2 ± 3.2 | 74.4 ± 1.5 | 45.5 ± 1.7 | 91.6 | 69.4 | 95.8 ± 4.3 | 97.5 ± 0.4 | 97.6 ± 0.6 | 68.9 | 80.8 |
| SSAM | 87.3 | 53.5 | 75.5 ± 1.4 | 82.9 ± 3.6 | 70.2 ± 0.4 | 56.4 ± 5.2 | 89.3 | 72.6 | 93.2 ± 3.5 | **97.8** ± 0.5 | 96.9 ± 0.5 | 69.0 | 79.6 |
| GRAM-Binaural | **95.6** | 56.1 | 81.0 ± 1.1 | 86.7 ± 2.4 | 75.0 ± 1.4 | 53.2 ± 3.0 | 92.5 | **77.0** | 94.9 ± 3.2 | 97.3 ± 0.3 | 98.1 ± 0.2 | 72.3 | 82.5 |
| GRAM-Ambisonics | 94.3 | 53.0 | 79.4 ± 1.5 | 85.9 ± 1.5 | 71.9 ± 1.9 | 53.7 ± 1.2 | 89.6 | 73.8 | 94.9 ± 4.9 | 97.6 ± 0.5 | **98.5** ± 0.4 | 71.3 | 81.1 |
| GRAM-Clean | 95.3 | 56.8 | **81.3** ± 1.8 | **87.5** ± 2.3 | **75.1** ± 0.6 | 57.3 ± 3.4 | 93.5 | 75.8 | 95.8 ± 3.7 | 97.4 ± 0.3 | 98.0 ± 0.2 | **73.8** | **83.1** |
| **Supervised** | | | | | | | | | | | | | |
| PASST | – | **64.1** | 60.7 ± 3.7 | **94.8** ± 0.3 | 61.8 ± 1.1 | 25.9 ± 2.6 | 68.7 | 24.2 | **96.6** ± 3.2 | 96.4 ± 0.7 | 87.8 ± 1.2 | 46.2 | 68.1 |
| Spatial-AST | – | 54.7 | 72.6 ± 1.5 | 90.3 ± 1.7 | 62.2 ± 1.3 | 29.1 ± 1.9 | 80.6 | 69.8 | 96.2 ± 5.3 | 96.2 ± 0.4 | 94.6 ± 0.6 | 54.6 | 74.6 |

Table 8: Training details of the recent audio foundation models. We retrieve the numbers from the references where possible. Various works utilized various sizes of AudioSet. Therefore, we used the dataset size reported by the references to calculate the steps per epoch. For MW-MAE and SSAM we retrieved their dataset size from their corresponding code repository.

| Model | Batch Size | Epochs | Steps per Epoch | Input Length | Total Samples Seen |
|---|---|---|---|---|---|
| MW-MAE (Yadav et al., 2024) | 1024 | 100 | 1985 | N/A | 2s |
| GRAMs | 96 | N/A | N/A | 180,000 | ∼2s |
| Audio-MAE (Huang et al., 2022) | 512 | 32 | 3829 | N/A | 10s |
| BEATs (Chen et al., 2023) | 5600 | N/A | N/A | 1.2M | 10s |
| SSAM (Yadav & Tan, 2024) | 1024 | 100 | 2003 | N/A | 2s |

## I    EVALUATING TRAINING EFFICIENCY

For all models trained solely on AudioSet, we calculated the number of seconds seen during the training as: batch size × steps per epoch × epochs × input length. This comparison accounts for the number of 10-second AudioSet sound clips processed by each model.

