# OpenReview forum: "GRAM: Spatial general-purpose audio representations for real-world applications"
_ICLR.cc/2026/Conference — Submitted to ICLR 2026_

### Official Review · Reviewer_rz9E · 2025-10-31

**Soundness:** 2
**Presentation:** 3
**Contribution:** 2
**Rating:** 2
**Confidence:** 5

**Summary:**

The paper aims to provide a model that can handle spatial audio. To that end, a multi-channel masked auto-encoder is trained for two-channel and four-channel audio formats.

**Strengths:**

There have significant advances on foundation models for speech. However, spatial audio has been largely overlooked. It is positive to see advances in this area. To this end, the paper provides a potentially valuable dataset containing binaural and ambisonics room impulse responses. The paper also provides a reasonable evaluation of the proposed model for several audio datasets, compared to some alternative audio models.

**Weaknesses:**

I have two major concerns about the claims of this work: 1) A key challenge in spatial audio is the sheer variety of different microphone array geometries (ranging from binaural listeners to linear, circular, and spherical arrays of varying number of microphones). By definition, a foundation model needs to generalise across different array geometries. However, the proposed approach is limited to only two specific formats involving only two and four microphones. While the results are very promising for these specific formats, the model is not suitable for other, more general geometries. 2) The model is trained on spectrograms (or spectrograms + intensity vectors). Considering that spectrograms discard phase information (which is crucial for spatial audio), it is unclear why the model is not trained on raw waveforms, similar to recent foundation models for speech.

**Questions:**

-	Can the model be extended to more general microphone array geometries without the need for re-training?
-	How does the model perform when trained on raw waveforms rather than spectrograms?
-	How did you train the Mamba variant? Based on the discussion in the paper, I believe this trained using spectrograms as inputs. However, it is unclear why you would train a sequence model on image-like data.

---

> ### Author Response · Authors · 2025-11-25
> **Rebuttal 1/2**
>
> ## W1/Q1: Microphone Array Geometry Coverage
>
> We appreciate the reviewer's attention to generalizability across array geometries. We would like to address what we believe may be a misunderstanding about the scope of foundation models and clarify our motivation for GRAM-Binaural and GRAM Ambisonics.
>
> **Clarification on "Foundation Model" Terminology:**
>
> The term "audio foundation models" refers to models that learn general-purpose representations that can robustly perform a wide range of downstream tasks [1, 2], including environmental sound classification, sound scene analysis, keyword spotting, and music genre recognition. In recent years, several benchmark task suites have been introduced to evaluate the performance of such audio foundation models, such as HEAR [1], and HARES [2], to evaluate the general-purpose audio representation learning capabilities of such models. Examples of well-known audio foundation models are AST  [3], SSAST  [4], BEATs [5] , and  Audio-MAE [6].  These foundation models all undergo an extensive pre-training regime, albeit non-spatialized and non-reverberant. Later, embeddings extracted by these models are used for general-purpose downstream tasks with a linear head to learn task-specific mapping.  Thus, in the literature, the term audio **foundation** models refers to models with broad downstream applicability rather than models accommodating a wide range of microphone geometries.
>
> GRAM’s unique  contribution is the incorporation of spatial information directly into a unified embedding space, enabling both typical audio foundation model tasks and localization tasks from a single, general-purpose audio representation. This represents a significant advance, as previous models either addressed these tasks in a supervised manner, explicitly modeled spatial information [7], or relied on concatenating separate spectrotemporal and sound localization models [8]. GRAM instead achieves state-of-the-art performance on a wide range of downstream tasks in naturalistic listening scenes, including reverberation and background noise, by learning spatial general-purpose representations in a computationally efficient manner using the proposed multi-channel masked auto-encoder approach. Furthermore, naturalistic training not only enhances the performance in naturalistic scenes, but also in clean scenes without spatial and reverberant attributes.
>
> **Rationale for GRAM-Binaural and GRAM-Ambisonics:**
>
> GRAM aims to close the performance gap of audio foundation models in spatial, real-world listening scenes that include background noise and reverberation. We chose to develop two GRAM versions, GRAM-Binaural and GRAM-Ambisonics, to show that our proposed multi-channel masked auto-encoding approach learns robust spatial general-purpose audio representations from binaural audio clips (mimicking human hearing) as well as from the most commonly used spatial audio format, ambisonics. Ambisonics are the basis for research on spatial audio capture and reproduction [9].  Furthermore, deep learning with audio clips in ambisonics format has been repeatedly demonstrated to give superior results on localization tasks compared to other microphone arrays [10, 11, 12, 13]. Finally, it is worth noting that GRAM is the first audio foundation model to accommodate both binaural and ambisonic audio formats.
>
> In sum, while the multi-channel masked auto-encoder approach underlying GRAM can be extended to other microphone geometries, the actual implementation thereof is not within the scope of a typical audio foundation model paper, as foundation models focus on broad downstream applicability.

---

> ### Author Response · Authors · 2025-11-25
> **Rebuttal 2/2**
>
> ## W2/Q2: Training on Spectrograms vs. Raw Waveforms
>
> We thank the reviewer for pointing out the need for greater clarification regarding the motivation for a spectrogram-based rather than a waveform-based model.
>
> Firstly, while phase information is relevant for sound localization, sound location can also be derived from intensity differences (e.g., from interaural level differences in the binaural format and from intensity vectors in the ambisonics format). Our superior localization results highlight that phase information is not necessarily needed for capturing spatial attributes in a self-supervised manner and that spectrograms (GRAM-Binaural) and a combination of spectrograms and intensity vectors (GRAM-Ambisonics) carry rich enough spatial information.
> Furthermore, we chose to use a spectrogram-based approach because the primary goal of GRAM is to bridge the performance gap of audio foundation models in spatial, real-world listening scenarios as efficiently as possible for general-purpose representation learning. In particular, log-mel spectrograms provide a more compact representation compared to very high-dimensional raw waveforms and allow the model to learn both the temporal and frequency structure [14]. Therefore, recent methods extensively utilize log-mel features [4,5,6,7,14], achieving impressive results on general-purpose representation learning. Whereas methods with raw waveform have only been shown to perform well on speech-related tasks [1, 2].
>
> Moreover, we would like to point out that waveform-based speech models (as indicated by the reviewer) still perform poorly on general-purpose tasks even when trained on AudioSet and are thus not suitable candidates for the current aims. Table 1 and Table 2 represent the results of Wav2Vec2.0 [15] and HuBERT [16] models pre-trained on AudioSet (Models taken from [17])  compared to GRAMs.
>
> Table 1: HEAR performance of AudioSet pre-trained Wav2Vec2.0 (Base) and HuBERT (Base) compared to GRAMs
> | Model | DCASE | FSD50K | LC | ESC-50 | CD |VL | SC-5 | NS | BO | Mri-S | Mri-T |
> | -- | -- | -- | -- | -- | -- | -- | -- | -- | -- | -- | -- |
> Wav2Vec2.0|  52.0 | 34.7 | 60.4 ± 1.7 | 58.9 ± 1.9 | 56.3 ± 1.3 | 27.9 ± 4.6 | 72.1 |42.0 |86.0 ± 9.6 | 92.9 ± 1.4 | 77.3 ± 0.5
> HuBERT | 86.2 | 41.1 | 63.5 ± 3.4 | 69.1 ± 1.6 | 69.5 ± 1.2 | 53.3 ± 3.1 | 83.5 | 38.8 | 91.5 ± 8.8 | 95.6 ± 0.5 | 90.4 ± 0.8
> GRAM-Binaural | 95.6 | 56.1 | 81.0 ± 1.1 | 86.7 ± 2.4 | 75.0 ± 1.4 | 53.2 ± 3.0 | 92.5 | 77.0 | 94.9 ± 3.2 | 97.3 ± 0.3 | 98.1 ± 0.2
> GRAM-Ambisonics | 94.3 | 53.0 |79.4 ± 1.5 | 85.9 ± 1.5 | 71.9 ± 1.9 | 53.7 ± 1.2 | 89.6 | 73.8 | 94.9 ± 4.9 | 97.6 ± 0.5 | 98.5 ± 0.4
> GRAM-Clean | 95.3 | 56.8 | 81.3 ± 1.8 | 87.5 ± 2.3 | 75.1 ± 0.6 | 57.3 ± 3.4 | 93.5 | 75.8 | 95.8 ± 3.7 | 97.4 ± 0.3 | 98.0 ± 0.2
>
> Table 2: Nat-HEAR performance of AudioSet pre-trained Wav2Vec2.0 (Base) and HuBERT (Base) compared to GRAMs
> | Model | DCASE | FSD50K | LC | ESC-50 | CD |VL | SC-5 | NS | BO | Mri-S | Mri-T |
> | -- | -- | -- | -- | -- | -- | -- | -- | -- | -- | -- | -- |
> Wav2Vec2.0 | 33.1 |27.7 |51.0 ± 1.2 | 48.1 ± 2.1 | 43.9 ± 2.2 | 22.3 ± 1.5 | 60.1 | 21.2 | 75.8 ± 6.0 | 74.4 ± 1.6 | 45.2 ± 1.5
> HuBERT | 69.8 |34.7 | 53.0 ± 1.0 | 56.6 ± 2.5 | 48.9 ± 1.6 | 40.6 ± 2.0 | 76.3 | 29.8 | 80.1 ± 5.8 | 79.3 ± 1.1 | 52.8 ± 1.2
> GRAM-Binaural | 93.0 | 52.8 | 72.3 ±0.7 | 82.6 ± 3.2 | 63.3 ± 1.3 | 35.1 ± 3.8 | 91.0| 67.6 | 85.6 ± 5.1 | 91.7 ± 0.9 | 78.3 ± 1.3
> GRAM-Ambisonics | 90.2 | 49.5 | 68.8 ± 0.9 | 79.4 ± 2.7 | 61.4 ± 0.9 | 36.4 ± 4.2 | 87.2 | 64.6 | 83.4 ± 4.7 | 91.3 ± 0.6| 78.1 ± 1.4
> GRAM-Clean | 90.9 | 50.5 | 66.4 ± 0.8 | 80.0 ± 2.4 | 62.0 ± 1.3 | 32.2 ± 2.3 | 87.3 | 65.2 | 82.2 ± 5.6 | 90.2 ± 0.8 | 75.1 ± 0.7
>
>
>
> ## Q3: Mamba Variant Training
>
> We trained the Mamba variant following established practices from [18,19]. While the reviewer notes that spectrograms are "image-like," and Mamba would not be a suitable choice for modelling images, it is well-documented that Mamba architectures successfully apply to image domains and achieve strong performance on vision tasks [20].

---

> ### Author Response · Authors · 2025-11-25
> **References 1/2**
>
> [1] Turian, J., Shier, J., Khan, H. R., Raj, B., Schuller, B. W., Steinmetz, C. J., Malloy, C., Tzanetakis, G., Velarde, G., McNally, K., Henry, M., Pinto, N., Noufi, C., Clough, C., Herremans, D., Fonseca, E., Engel, J., Salamon, J., Esling, P., … Bisk, Y. (2022, March 6). HEAR: Holistic Evaluation of Audio Representations. arXiv.Org. https://arxiv.org/abs/2203.03022v3
>
> [2] Wang, L., Luc, P., Wu, Y., Recasens, A., Smaira, L., Brock, A., Jaegle, A., Alayrac, J.-B., Dieleman, S., Carreira, J., & Oord, A. van den. (2021, November 23). Towards Learning Universal Audio Representations. arXiv.Org. https://arxiv.org/abs/2111.12124v3
>
> [3] Gong, Y., Chung, Y.-A., & Glass, J. (2021). AST: Audio Spectrogram Transformer (No. arXiv:2104.01778). arXiv. https://doi.org/10.48550/arXiv.2104.01778
>
> [4] Gong, Y., Lai, C.-I. J., Chung, Y.-A., & Glass, J. (2021, October 19). SSAST: Self-Supervised Audio Spectrogram Transformer. arXiv.Org. https://arxiv.org/abs/2110.09784v2
>
> [5] Chen, S., Wu, Y., Wang, C., Liu, S., Tompkins, D., Chen, Z., & Wei, F. (2022, December 18). BEATs: Audio Pre-Training with Acoustic Tokenizers. arXiv.Org. https://arxiv.org/abs/2212.09058v1
>
> [6] Huang, P.-Y., Xu, H., Li, J., Baevski, A., Auli, M., Galuba, W., Metze, F., & Feichtenhofer, C. (2022, July 13). Masked Autoencoders that Listen. arXiv.Org. https://arxiv.org/abs/2207.06405v3
>
> [7] Zheng, Z., Peng, P., Ma, Z., Chen, X., Choi, E., & Harwath, D. (2025). BAT: Learning to Reason about Spatial Sounds with Large Language Models (No. arXiv:2402.01591). arXiv. https://doi.org/10.48550/arXiv.2402.01591
>
> [8] Devnani, B., Seto, S., Aldeneh, Z., Toso, A., Menyaylenko, E., Theobald, B.-J., Sheaffer, J., & Sarabia, M. (2024). Learning Spatially-Aware Language and Audio Embeddings (No. arXiv:2409.11369). arXiv. https://doi.org/10.48550/arXiv.2409.11369
>
> [9] Zhang, W., Samarasinghe, P. N., Chen, H., & Abhayapala, T. D. (2017). Surround by Sound: A Review of Spatial Audio Recording and Reproduction. Applied Sciences, 7(5), 532. https://doi.org/10.3390/app7050532
>
> [10] Lan, C., Zhang, L., Zhang, Y., Fu, L., Sun, C., Han, Y., & Zhang, M. (2022). Attention mechanism combined with residual recurrent neural network for sound event detection and localization. EURASIP Journal on Audio, Speech, and Music Processing, 2022(1), 29. https://doi.org/10.1186/s13636-022-00263-6
>
> [11] Nguyen, T. N. T., Watcharasupat, K. N., Nguyen, N. K., Jones, D. L., & Gan, W.-S. (2022). SALSA: Spatial Cue-Augmented Log-Spectrogram Features for Polyphonic Sound Event Localization and Detection. IEEE/ACM Transactions on Audio, Speech, and Language Processing, 30, 1749–1762. https://doi.org/10.1109/TASLP.2022.3173054
>
> [12] Politis, A., Shimada, K., Sudarsanam, P., Adavanne, S., Krause, D., Koyama, Y., Takahashi, N., Takahashi, S., Mitsufuji, Y., & Virtanen, T. (2022). STARSS22: A dataset of spatial recordings of real scenes with spatiotemporal annotations of sound events (No. arXiv:2206.01948). arXiv. https://doi.org/10.48550/arXiv.2206.01948
>
> [13] Shimada, K., Politis, A., Sudarsanam, P., Krause, D., Uchida, K., Adavanne, S., Hakala, A., Koyama, Y., Takahashi, N., Takahashi, S., Virtanen, T., & Mitsufuji, Y. (2023). STARSS23: An Audio-Visual Dataset of Spatial Recordings of Real Scenes with Spatiotemporal Annotations of Sound Events (No. arXiv:2306.09126). arXiv. https://doi.org/10.48550/arXiv.2306.09126
>
> [14] Niizumi, D., Takeuchi, D., Ohishi, Y., Harada, N., & Kashino, K. (2022). Masked Spectrogram Modeling using Masked Autoencoders for Learning General-purpose Audio Representation. HEAR: Holistic Evaluation of Audio Representations, 1–24. https://proceedings.mlr.press/v166/niizumi22a.html

---

> ### Author Response · Authors · 2025-11-25
> **References 2/2**
>
> [15] Baevski, A., Zhou, H., Mohamed, A., & Auli, M. (2020). wav2vec 2.0: A Framework for Self-Supervised Learning of Speech Representations (No. arXiv:2006.11477). arXiv. https://doi.org/10.48550/arXiv.2006.11477
>
> [16] Hsu, W.-N., Bolte, B., Tsai, Y.-H. H., Lakhotia, K., Salakhutdinov, R., & Mohamed, A. (2021). HuBERT: Self-Supervised Speech Representation Learning by Masked Prediction of Hidden Units (No. arXiv:2106.07447). arXiv. https://doi.org/10.48550/arXiv.2106.07447
>
> [17] Quatra, M. L., Koudounas, A., Vaiani, L., Baralis, E., Cagliero, L., Garza, P., & Siniscalchi, S. M. (2024). Benchmarking Representations for Speech, Music, and Acoustic Events. 2024 IEEE International Conference on Acoustics, Speech, and Signal Processing Workshops (ICASSPW), 505–509. https://doi.org/10.1109/ICASSPW62465.2024.10625960
>
> [18] Shams, S., Dindar, S. S., Jiang, X., & Mesgarani, N. (2024). SSAMBA: Self-Supervised Audio Representation Learning with Mamba State Space Model. 2024 IEEE Spoken Language Technology Workshop (SLT), 1053–1059. https://doi.org/10.1109/SLT61566.2024.10832304
>
> [19] Yadav, S., & Tan, Z.-H. (2024). Audio Mamba: Selective State Spaces for Self-Supervised Audio Representations (No. arXiv:2406.02178). arXiv. https://doi.org/10.48550/arXiv.2406.02178
>
> [20] Zhu, L., Liao, B., Zhang, Q., Wang, X., Liu, W., & Wang, X. (2024). Vision Mamba: Efficient Visual Representation Learning with Bidirectional State Space Model (No. arXiv:2401.09417). arXiv. https://doi.org/10.48550/arXiv.2401.09417

---

> ### Comment · Reviewer_rz9E · 2025-11-27
> **Response to Rebuttal 1/2**
>
> Thank you for the detailed rebuttal.
>
> ## Foundation models
> Foundation models refer to models that are “trained on broad data at scale and are adaptable to a wide range of downstream tasks” as defined in [Bommasani et al, 2021]. My concern is the lack of *broad data*:
>
> Existing models - including, for example AST, SSAST, BEATS and Audio-MAE - capture “broad data” by training on a large variety of sound events and recording conditions (e.g., AudioSet, HEAR). This is suitable for audio models that are not concerned with spatial cues.
>
> In contrast to mono audio, one of the key challenges for spatial audio is that different devices have vastly different microphone array geometries. The geometry (binaural, linear, planar, circular, spherical, etc) and configuration (number of mics, inter-mic distance, directivity patterns, etc) of microphone arrays impacts substantially on the information that is embedded in the array signals (e.g., aliasing, angular sensitivity). For example, localisation performance can vary drastically with the inter-microphone distance in an array.
>
> A general purpose, foundation model, needs to be able to handle these differences as these are core challenges of spatial audio.
>
> ## Audio formats
> I appreciate that ambisonics and binaural formats are common formats for specialist areas in the audio community (e.g., sound field reproduction). However, my concern is their relevance to the general machine learning community at ICLR, where researchers are likely to be more interested in arrays that are integrated in devices such as smartphones, virtual reality headset, or voice assistants. Microphone arrays in these devices are often either linear, circular or planar. To the best of my knowledge, a 3D microphone array is required to compute first-order ambisonics.
>
> **Note:** The model is constrained to first-order ambisonics, which has well-known limitations, e.g., performance at high frequencies.
>
> ## Spectrograms
> My concern here is the scope of tasks, given the limitation to spectrograms. This comes back to the definition of foundation models: generality across a *wide range* of downstream tasks. The paper presents promising results on downstream tasks such as sound event detection, source localization and T60 estimation. However, relevant audio applications require models that can predict signals that human end-users can listen to (rather than visual representations, such as spectrograms), for example for spatial audio rendering, dereverberation. It is well-known in the audio community that waveform reconstruction from spectrograms is problematic due to missing phase signals. Therefore, complex scaffolding would be required around GRAM to handle the missing phase, including, for example vocoders [Lee 2023], [Kong 2020].
>
> While GRAM achieves promising results for discriminative downstream tasks (localization, T60 estimation), I am concerned that the use of log-melspectra limits its usefulness for more general spatial audio applications.
>
> **Note:** I am not convinced that your current experiments substantiate the claim that “Our superior localization results highlight that phase information is not necessarily needed”. I agree with reviewer zoEh that it would be interesting to include a comparison of the ambisonics variant with and without intensity vectors.
>
> ## Mamba
> Transforming inherently sequential data into image-like representations (spectrograms) only to subsequently convert them back into sequences using Mamba seems conceptually incoherent and computationally inefficient. Further, spectrograms are **not** natural images and the correlation between time-frequency bins in spectrograms is very different from the relationship between pixels in natural images.
>
> It remains unclear how Mamba is actually applied. The paper states that “we trained also an encoder with a state-of-the-art 8-layer Mamba architecture (Gu & Dao, 2023; Yadav & Tan, 2024)”? Vanilla mamba often unrolls images as a 1D sequence. However, this 1D unrolling destroys 2D structure in spatial data for images, see, e.g., [Zhang 2025]. I would expect similarly adverse effects for spectral-temporal representations in audio.
>
> Unfortunately, you did not answer my question and the paper does not provide information about the unrolling. Without a clear description of how 2D spectral-temporal signals are handled, it is difficult to draw any meaningful conclusions from your Mamba results.
>
> ## Overall
> I believe that the idea of a foundation model for spatial audio is exciting and has potential to make impact. However, in my opinion, core challenges of foundation models for spatial audio remain unaddressed, and GRAM's input formats / representations (limited to binaural and ambisonics formats; spectrogram inputs) limit its contribution.

---

> ### Comment · Reviewer_rz9E · 2025-11-27
> **Response to Rebuttal 2/2**
>
> ## References
> [Bommasani 2021] Bommasani et al., “On the Opportunities and Risks of Foundation Models,” https://crfm.stanford.edu/assets/report.pdf
>
> [Kong 2020] Kong, Kim & Bae, “HiFi-GAN: Generative Adversarial Networks for Efficient and High Fidelity Speech Synthesis,” NeurIPS 2020 (https://arxiv.org/abs/2010.05646)
>
> [Lee 2023] Lee et al., “BigVGAN: A Universal Neural Vocoder with Large-Scale Training”, ICLR 2023
>
> [Zhang 2025] Zhang et al., "2DMamba: Efficient State Space Model for Image Representation with
> Applications on Giga-Pixel Whole Slide Image Classification", CVPR 2025 (https://openaccess.thecvf.com/content/CVPR2025/papers/Zhang_2DMamba_Efficient_State_Space_Model_for_Image_Representation_with_Applications_CVPR_2025_paper.pdf)

---

> > ### Author Response · Authors · 2025-12-03
> >
> > **Foundation models and audio formats:**
> >
> > We understand the viewpoint of the reviewer about the term “foundation model” from the perspective of a general AI audience, as is indeed the case at ICLR. We would like to mention two points.
> >
> > * The term foundation model is used differently in the field of audio AI: The papers referenced by the reviewer actually refer to themselves as foundation models, even though these are single-channel models that cannot deal with multi-channel audio at all, focusing instead on the broad range of downstream tasks as we do in the paper. However, the term foundation model is not a critical component for our work as the main aim is (as reflected by the title of the manuscript) to learn a spatial general-purpose audio representation. The term foundation model could therefore also be removed, focusing instead only on “spatial general-purpose audio representation”.
> >
> > *We would like to argue that not including all microphone geometries is not a limiting factor in our work, as our work shows for the very first time the potential of learning powerful, general-purpose spatial audio representations for two different (and commonly used) microphone geometries, and, crucially, GRAM can be trained on any microphone geometry. The approach is thus applicable to any type of microphone geometry/audio format, as we demonstrate for two widespread geometries. Moreover, these types of models – even without a spatial component – receive widespread attention and citations (see the references by the reviewer), suggesting that there is a wide interest in this topic of spatial general-purpose audio representation and that our paper will have a significant impact on the development of this field.
> >
> > * Finally, while it is true that ambisonics require a 3D microphone format, we demonstrate the applicability of GRAM to e.g. linear arrays with the binaural format, which is in essence a two-channel linear array. Moreover, one very prominent application would be assistive hearing devices, which indeed directly translate to the binaural two-channel format presented here. Extending to other formats (e.g. multi-channel derived from a circular microphone array) can be easily done.
> >
> >
> >
> > **Spectrograms**
> >
> > We agree with the reviewer that the use of spectrograms imposes limitations on the output quality for generative audio tasks such as speaker separation or dereverberation. However, while we plan as next steps to develop a generative pipeline as well (as indicated in the manuscript also), we selected a spectrogram approach here to mitigate the impact of the distortions introduced by background noise and reverberations. Our results show that this is successful, and we would like to point out that our approach outperforms all time-domain-based models. Moreover, we would like to emphasize that all models referred to by the reviewer that are listed as “audio foundation models” are in fact spectrogram-based models like GRAM. It is thus not clear what the motivation is for considering the use of spectrograms for general-purpose spatial audio representations void for GRAM, even though this is widely accepted in the field.
> >
> > Intensity vectors:
> >
> > As per the request of the reviewers, we ran the proposed experiment. We find that intensity vectors are crucial for learning spatial attributes for ambisonics (as expected, mel spectrograms alone do not carry enough spatial information). Whereas for binaural inputs, the mel spectrograms carry the ILD information, and thus, localization can be done without the phase encoding.
> >
> > Table 1: HEAR Results
> > |Model|ESC-50|LC|CD|VL
> > |---|---|---|---|---|
> > |GRAM-Ambisonics Mel |81.6 +- 3.3 |77.5 +- 1.3 |51.5 +- 11.1 |42.5+- 3.7|
> > |GRAM-Ambisonics Mel + Active|82.3 +- 1.9 |78.2 +- 1.0 |70.3 +- 0.9| 43.8 +- 2.7
> > |GRAM-Ambisonics Mel + Active + Reactive|78.5 +- 2.5 |77.0 +- 1.6|70.0 +- 1.1| 41.8 +- 2.9
> >
> > Table 2: NatHEAR Results
> > |Model|ESC-50|LC|CD|VL
> > |---|---|---|---|---|
> > |GRAM-Ambisonics Mel |77.1 +- 2.9|68.5 +- 1.4|56.5 +- 0.8|76.6 +- 1.1|
> > |GRAM-Ambisonics Mel + Active|74.5 +- 2.2|71.9 +- 1.2|56.4 +- 0.8 |26.8 +- 2.7
> > |GRAM-Ambisonics Mel + Active + Reactive|72.0 +- 1.6|75.4 +- 1.9|55.2 +- 0.9 |25.9 +- 2.6
> >
> > Table 3: Localization Results
> > |Model|ESC-50|SC
> > |---|---|---|
> > |GRAM-Ambisonics Mel |82.6|69.1
> > |GRAM-Ambisonics Mel + Active|23.4|15.2
> > |GRAM-Ambisonics Mel + Active + Reactive|23.9|17.8
> >
> >
> > Mamba
> >
> > We apologize for the confusion. We followed the exact training methodology of [1]. We split the input spectrogram into non-overlapping patches. We then flatten these patches and project them linearly to followed by adding
> > fixed sinusoidal positional embeddings for encoding 2D positional information.
> >
> > [1] Yadav, S., & Tan, Z.-H. (2024). Audio Mamba: Selective State Spaces for Self-Supervised Audio Representations (No. arXiv:2406.02178). arXiv. https://doi.org/10.48550/arXiv.2406.02178

---

### Official Review · Reviewer_ESjb · 2025-10-31

**Soundness:** 3
**Presentation:** 3
**Contribution:** 2
**Rating:** 4
**Confidence:** 2

**Summary:**

The paper presents GRAM (General-purpose, Real-world Audio Model), a multi-channel masked autoencoder (MAE) designed to learn spatially aware audio representations from simulated naturalistic sound scenes. Unlike existing single-channel foundation models such as Audio-MAE or BEATs, GRAM explicitly encodes spatial cues by processing both binaural (2-channel) and Ambisonics (4-channel) inputs. To support systematic evaluation, the authors introduce Nat-HEAR, an extended version of the HEAR benchmark that incorporates spatialized and localization-oriented downstream tasks. Experimental results show that GRAM achieves state-of-the-art performance across multiple baselines, both self-supervised and supervised, and demonstrates notable robustness to reverberation and environmental noise. Together, GRAM and Nat-HEAR represent an effort to push foundation audio models toward more realistic spatial understanding.

**Strengths:**

(1) The application of a masked autoencoder to multi-channel spectrograms is an important step toward spatially aware audio modeling. By allowing the network to learn from interaural time and intensity differences as well as room reverberation cues, the approach extends the MAE framework beyond monaural “dry” signals to capture the physical geometry of sound propagation. This opens the door to richer and more perceptually grounded audio representations.
(2) The introduction of Nat-HEAR as a benchmark is a strong contribution to the field. By expanding HEAR to include spatialized datasets and localization-oriented downstream tasks, the paper provides a valuable resource for evaluating and comparing spatial audio foundation models. The release of 85,000 simulated room impulse responses (BRIRs/ARIRs) also contributes significantly to the reproducibility and standardization of future research in this area.
(3) The paper demonstrates consistent performance improvements over both self-supervised and supervised baselines. Of particular note is that GRAM surpasses some supervised localization models trained with explicit direction labels, suggesting that the model’s spatial embeddings capture meaningful physical relationships without the need for handcrafted supervision.
(4) The study includes thoughtful ablations on key design factors such as masking strategies (patch- versus time-based), ratios of dry to reverberant mixtures, and backbone architecture choices (Transformer versus Mamba). These analyses provide transparency and help readers understand how GRAM’s components influence its robustness and generalization behavior.

**Weaknesses:**

(1) The architectural novelty of GRAM is limited. The model primarily extends existing single-channel frameworks such as Audio-MAE or BEATs to handle multi-channel inputs, without introducing a fundamentally new objective, representation mechanism, or training paradigm. While the extension is useful, the contribution is incremental in nature rather than conceptually transformative.
(2) Despite claims of strong real-world generalization, the training and evaluation are predominantly conducted on simulated environments rather than real-world multi-channel recordings. Validation on measured datasets such as SONYC-UST, TAU Spatial Sound, or FAIR-Play would be necessary to substantiate the claim that GRAM transfers effectively to real acoustic conditions, including non-ideal sensor responses and background variability.
(3) The paper’s positioning relative to recent spatial audio foundation models is not sufficiently clarified. Prior works such as Spatial-AST (Zheng et al., 2024) and Qwen-Audio (Chu et al., 2023) already explore similar multi-channel and spatial reasoning concepts within transformer frameworks. GRAM does not convincingly demonstrate performance superiority or distinct methodological advances compared to these approaches, especially on real-world localization or speech-related tasks.
(4) The model’s treatment of multi-channel spectrograms as simple stacked image-like inputs misses opportunities for deeper spatial modeling. Unlike methods such as Spatial-SSL (2024) or SoundFieldNet, GRAM does not explicitly incorporate spatial phase correlation, coherence constraints, or directional energy cues, which are physically meaningful aspects of spatial sound. This omission limits both interpretability and potential generalization in physically complex environments.

**Questions:**

(1) Can the authors provide quantitative results demonstrating GRAM’s performance on real-world multi-channel datasets to validate its generalization beyond simulated environments and strengthen the claim of real-world robustness?
(2) Can the authors present a direct comparison or ablation against recent spatial audio foundation using a consistent evaluation setup to objectively demonstrate where GRAM offers performance or efficiency advantages, thereby clarifying its distinct contribution?

---

> ### Author Response · Authors · 2025-11-13
> **References**
>
> Thank you for your comments on our paper. We are currently preparing responses to your questions, and we would like to respond to your comment regarding SoundFieldNet, and Spatial-SSL as well. However, we are unable to find these references. Would you mind sending them to us by replying to this comment. Thank you in advance!

---

> ### Author Response · Authors · 2025-11-24
> **Rebuttal 1/3**
>
> We thank the reviewer for taking the time to read and comment on our work, as well as for the suggestions and questions that significantly improved the quality and impact of our contribution. Below, we present point-by-point responses to the issues raised by the reviewer.
>
> **Weakness (1) Limited architectural novelty**: Here, we would like to clarify and emphasize the key methodological contributions that this paper makes to the community. In addition, we present new experiments and results that further strengthen the novelty of our work.
>
> • GRAM utilizes a **new framework for spatial audio representation learning in real-world scenes** which consists of a **novel data-generation pipeline** and **an innovative learning strategy combining multi-channel masking and multi-channel output representations to learn spatial representations from Ambisonics as well as binaural audio formats elegantly and efficiently** (see point-by-point answers to Questions for details). Although we indeed build on existing masked autoencoder approaches such as MWMAE, note that the novelty of our approach lies simply and efficiently in that we combine multi-channel masking and multi-channel output representations to efficiently learn both spectrotemporal and spatial information **without utilizing explicit spatial cues, new loss functions, or more elaborate training schemes** that would reduce efficiency. As such, our approach clearly stands apart from other approaches utilizing, for example, multiple encoders for spatial and audio learning or explicitly modeling spatial cues or training with a supervised spatial task. Figure 2A illustrates the efficiency of GRAM in comparison to existing models such as MWMAE, MAE, and BEATs.
>
> We achieve this by elegantly combining multi-channel masking with multi-channel output reconstruction: Masks are applied binaurally (or across 7 channels for the new Ambisonics implementation), and output representations are binaural (or 4-channel). By calculating the MSE loss over the multi-channel output representations, we encourage GRAM to reproduce the inter-channel differences in the output extracted with the multi-channel masking strategy. In this way, the model learns both spectrotemporal information and spatial cues from the inter-channel comparisons in an elegant manner which greatly improves model efficiency. Our approach thus presents a straightforward, effective method to encourage masked autoencoder models (irrespective of the encoder architecture or audio format) to learn spatial representations from binaural and Ambisonics data.
>
> In sum, GRAM distinguishes itself from other spatial audio representation modelssuch as ELSA, SELDnet, and Spatial-AST, by avoiding the use of inefficient strategies such as multiple encoders or explicit loss terms. Moreover, the state-of-the-art performance of GRAM shows on naturalistic sound scenes and real-world recordings is unprecedented, highlighting the novelty and innovativeness of our approach.
>
> • We **assessed the generalization of the proposed multi-channel masked autoencoder approach across different network architectures** by directly comparing state-of-the-art Mamba and Transformer encoders, providing novel and highly relevant insights for the community.
>
> • GRAM is the first self-supervised spatial audio foundation model that successfully accommodates both binaural and Ambisonics data formats. GRAM is thus highly flexible and a valuable contribution for both technical and research applications

---

> ### Author Response · Authors · 2025-11-24
> **Rebuttal 2/3**
>
> **Q1**: We thank the reviewer for pointing out the need for further assessment of the real-world robustness and transferability of GRAM’s general-purpose audio representations. In agreement with this suggestion, we evaluated GRAM-Ambisonics on the challenging STARSS’23 [1] dataset, comprised of real-world recordings of multi-source sound scenes. While of interest, we did not include the FAIR-Play and SoNYC-UST suggested by the reviewer because STARSS23 already consists of overlapping sounds, interfering noise, localization labels, and realistic recording scenarios. Whereas FAIR-Play and SoNYC-UST lack localization labels.
>
> We assessed the transferability of GRAM-Ambisonics using both the linear-head approach as used for the Nat-HEAR benchmark tasks and a more extensive full-model fine-tuning procedure to accommodate the differences in data. Following the STARSS [1] convention, we reported at ER20◦,F20◦, LR_CD, and LE_CD. Two metrics are referred to as location-aware detection and are the error rate (ER20◦ ) and F-score (F20◦ ) in one-second non-overlapping segments. LE_CD expresses the average angular difference between the same class’s predictions and references. LR_CD tells the true positive rate of how many of these localization estimates were detected in a class out of the total number of class instances.
>
> As expected based on the different nature of the STARSS’23 data, the direct transfer of GRAM-Ambisonics representations using a linear-head approach results in poor performance (localization error of 47.1 °, Table 1). However, it should be noted that the poor performance was mainly caused by a failure to detect the event classes ‘door’ and ‘bell’ (F-score 0.0, Table 2), which automatically results in a localization error of 180°. Removing these classes reveals an average localization error of 22.9°. This suggests that the model's representations are effective for a majority of the classes and our pre-trained representations transfer accurately to challenging real-life recordings.
>
> To accommodate the different nature of the STARSS’23 dataset, we also investigated full model fine-tuning on STARSS’23 to allow the model to adapt to its unique challenges. As shown in Table 1, fine-tuning yields a dramatic improvement, bringing the LE_CD down to 18.6° and closing the performance gap considerably
> Furthermore, the fine-tuning dynamics illustrated in Figure 4 in our updated manuscript demonstrate that our pre-trained representations enable rapid and effective adaptation to the real world. The fine-tuned model achieves high performance within just 20 epochs, significantly outperforming a randomly initialized model trained from scratch, which plateaus at a lower performance level regardless of hyperparameter tuning. This demonstrates that our pre-training provides a robust and transferable initialization for complex, real-world tasks. We will include the STARSS’23 evaluation and results in the camera-ready manuscript.
>
> Table 1: STARSS23 Results, State-of-the-art (SOTA) model is taken from [2], Baseline is taken from [3].
>
> | Model                    | ER₂₀° ↓ | F₂₀° ↑ | LE_CD ↓ | LR_CD ↑ |
> | ------------------------ | ------- | ------ | ------- | ------- |
> | **Native Sampling Rate** |         |        |         |         |
> | SOTA                     | 0.42    | 59.0%  | 13°     | 72.0%   |
> | Baseline                 | 0.57    | 29.9%  | 22°     | 47.7%   |
> | **Upsampled**            |         |        |         |         |
> | GRAM-Amb. (Full model fine-tuning)    | 0.51    | 41.4%  | 18.6°   | 60.5%   |
> | GRAM-Amb. (Linear head)      | 0.59    | 23.8%  | 47.1°   | 35.0%   |
> | Baseline (Reprod.)       | 0.62    | 28.3%  | 23.7°   | 45.7%   |
>
>
> [1] Shimada et al (2023) "STARSS23: An Audio-Visual Dataset of Spatial Recordings of Real Scenes with Spatiotemporal Annotations of Sound Events" NeurIPS Track on Datasets and Benchmarks.
>
> [2] Qing Wang, Ya Jiang, Shi Cheng, Maocheng Hu, Zhaoxu Nian, Pengfei Hu, Zeyan Liu, Yuxuan Dong, Mingqi Cai, Jun Du, and Chin-Hui Lee. The nerc-slip system for sound event localization and detection of dcase2023 challenge. Technical report, DCASE2023 Challenge, June 2023
>
> [3] Kazuki Shimada, Archontis Politis, Parthasaarathy Sudarsanam, Daniel Krause, Kengo Uchida, Sharath Adavanne, Aapo Hakala, Yuichiro Koyama, Naoya Takahashi, Shusuke Takahashi, Tuomas

---

> ### Author Response · Authors · 2025-11-24
> **Rebuttal 3/3**
>
> Table 2: Class-wise metrics for GRAM-Ambisonics Linear head
> | Class | ER₂₀° ↓ | F₂₀° ↑ | LE_CD ↓| 	LR_CD ↑                          |
> |-------|-------------------|--------------------|-------------------------|--------------------|
> | 0     | 0.59 [0.56, 0.64] | 0.52 [0.40, 0.64]  | 25.27 [15.50, 34.99]    | 0.82 [0.78, 0.86]  |
> | 1     | 0.59 [0.56, 0.64] | 0.63 [0.56, 0.70]  | 15.17 [13.77, 16.61]    | 0.74 [0.70, 0.78]  |
> | 2     | 0.59 [0.56, 0.64] | 0.20 [0.03, 0.37]  | 18.19 [13.20, 23.17]    | 0.19 [0.09, 0.28]  |
> | 3     | 0.59 [0.56, 0.64] | 0.01 [-0.01, 0.03] | 36.21 [27.53, 48.90]    | 0.37 [0.06, 0.71]  |
> | 4     | 0.59 [0.56, 0.64] | 0.36 [0.28, 0.44]  | 17.63 [13.83, 21.18]    | 0.40 [0.31, 0.48]  |
> | 5     | 0.59 [0.56, 0.64] | 0.30 [0.16, 0.45]  | 20.54 [16.75, 24.11]    | 0.43 [0.23, 0.65]  |
> | 6     | 0.59 [0.56, 0.64] | 0.01 [-0.00, 0.02] | 37.45 [27.60, 48.61]    | 0.05 [0.02, 0.10]  |
> | 7     | 0.59 [0.56, 0.64] | 0.00 [0.00, 0.00]  | 180.00 [180.00, 180.00] | 0.00 [0.00, 0.00]  |
> | 8     | 0.59 [0.56, 0.64] | 0.24 [0.11, 0.39]  | 37.12 [22.14, 51.88]    | 0.63 [0.51, 0.73]  |
> | 9     | 0.59 [0.56, 0.64] | 0.40 [0.31, 0.50]  | 20.91 [18.99, 22.74]    | 0.66 [0.56, 0.78]  |
> | 10    | 0.59 [0.56, 0.64] | 0.28 [0.04, 0.50]  | 13.19 [8.69, 20.37]     | 0.19 [-0.02, 0.38]
> | 11    | 0.59 [0.56, 0.64] | 0.00 [0.00, 0.00]  | 180.00 [180.00, 180.00] | 0.00 [0.00, 0.00]  |
> | 12    | 0.59 [0.56, 0.64] | 0.13 [-0.14, 0.43] | 10.31 [-490.76, 176.36] | 0.07 [-0.08, 0.22] |
>
> **Weakness (3) Positioning with respect to recent spatial audio foundation models,Q2**: To contextualize the performance of our self-supervised GRAM models, we extended the HEAR and NatHEAR benchmarks with two baselines: **Spatial-AST**(trained on spatialized AudioSet) and **PASST** (trained on AudioSet). The results are presented in Table 3.
> Table 3:
> |Model|NatHEAR Avg.|NatHEAR s(m)|HEAR Avg.|HEAR s(m)|
> |---|---|---|---|---|
> |GRAM-Binaural|**73.9**|**74.8**|82.5|72.3|
> |GRAM-Clean|71.1|67.3|**83.1**|**73.8**|
> |GRAM-Ambisonics|71.8|70.5|81.1|71.3|
> |Spatial-AST|52.8|30.9|74.6|54.6|
> |PASST|57.9|56.2|68.1|46.2|
>
> A clear trend emerges from the benchmarks: our self-supervised GRAMs consistently outperform both supervised baselines across nearly all tasks. It is noteworthy, however, that Spatial-AST and PASST demonstrate strong performance on specific tasks like ESC-50 and FSD50K. This is likely due to their supervised training on AudioSet, which shares a similar label distribution with these datasets, giving them an advantage in those specific domains.
>
> When compared directly to the most relevant baseline, Spatial-AST, our GRAM models demonstrate several key advantages:
>
> - **Superior Data Efficiency:** GRAMs achieve higher performance with less data, as evidenced by the learning curves in Figure 2A.
> - **Enhanced Localization Capabilities:** As shown in Figure 3A of the main paper, GRAMs learn more robust spatial representations.
> - **Stronger Overall Benchmark Performance:** GRAMs achieve higher aggregate scores on both the HEAR and NatHEAR benchmarks (Tables 3 & 4).
> - **Greater Robustness:** GRAMs are more resilient to challenging acoustic conditions like noise and reverberation (Figure 3B).
>
> Critically, GRAM-Binaural learns these powerful localization cues without explicit phase-based input or localization supervision, demonstrating its ability to discover useful spatial representations from audio alone. Finally, we highlight that GRAM-Ambisonics is, to our knowledge, the first Ambisonics-based foundation model capable of state-of-the-art general-purpose audio representation learning.
>
> Please note that Qwen-Audio consists of 670-million parameter audio encoder, and a 7B parameter large language model. Moreover, Qwen-Audio utilizes approximately 130k hours of pre-training dataset. Compared to our and other models trained on AudioSet (5k hours) therefore would not constitute as a fair comparison.
>
> **Weakness (4): Spatial modeling:** We would like to emphasize that our choice for modeling spatial encoding with multi-channel masking and reconstruction is made on purpose, as this presents a more efficient learning strategy than explicitly modeling spatial information. Furthermore, the presented approach is original and innovative, as explicit spatial modeling has been done previously by others, yet with substantially less success than spatial learning by GRAM (see also response to Weakness (1) above).
>
> Finally, we would like to point out that the papers referenced by the reviewer are not findable online. In order to take them into account, please provide the references (see also our communication of two weeks ago, November 13).

---

### Official Review · Reviewer_zoEh · 2025-11-03

**Soundness:** 3
**Presentation:** 1
**Contribution:** 2
**Rating:** 6
**Confidence:** 4

**Summary:**

This paper presents GRAM (general-purpose real-world audio model), a model which produces representations of spatial audio (from binaural or ambisonics inputs), capturing the main audio signal as well as other characteristics such as the reverberation, background noise, and spatial dimensions.

GRAM is trained as a masked auto-encoder, on a custom dataset with 85,000 room impulse responses (respectively for binaural and for ambisonics). Additionally, the authors extend the HEAR benchmark to incorporate the acoustic properties (resulting in Nat-HEAR).

Evaluations on the Nat-HEAR benchmark show GRAM-binaural achieving the highest combined score, followed by GRAM-Ambisonics. On the original HEAR benchmark, GRAM-clean (followed by GRAM-binaural, and GRAM-ambisonics) demonstrating GRAM's performance in natural sound-fields is not at the expense of performance on clean audio. Additionally, GRAM is show to capture directionality of sound with an 11.3° MAE on the TUT Sound Events 2018 REAL dataset.

**Strengths:**

* This paper tackles an important problem (representation learning for spatial audio) and does so thoroughly, ablating input format (binaural and ambisonics), architecture (transformer and mamba), patching strategy (frequency based and time based), masking ratio, and in-batch sampling factor.
* The paper is well written and easy to follow (though see a few questions below)

**Weaknesses:**

* **W1**: Though GRAM includes extensive ablations on Nat-HEAR and HEAR (which measure the understanding of the audio signal) as well as localisation tasks, its impact would be further strengthened with other downstream tasks (particularly around acoustics). Examples of other acoustics tasks include: source separation (as the authors suggest in the introduction) or T60 estimation.


* **W2**: There are some readability issues in the paper that need addressing (see questions below).

**Questions:**

* **Q1**: In line 225, if the hop window is 10ms, and the audio duration is 10s, why is the dimensionality of the spectrogram 1024x128 instead of 1000x128?
* **Q2**: If I understand in-batch sampling correctly, it implies that for a sample it gets repeated 3.2 times on average in a single batch. That could give raise to a very noisy gradient. Could you run pre-training with a sampling factor of 1 and use batch accumulation of 16 (so that you get the same effective batch size but with more diverse samples? I understand this might be up to 16x times slower to train, but that may be offset by faster convergence.
* **Q3**: GRAM-clean is not mentioned before line 251, and it is not clarified exactly what it does. I assume it's a variant of GRAM with the same Mel-spectrogram but using only a single channel?
* **Q4**: The average score for GRAM-Binaural does not seem to add up. I'm not sure I follow the computation of $s(m)$. Would it not be simpler to take the average performance? Or the average of the improvements over the baseline?
* **Q5**: [Minor] Note that TUT Sound Events 2018 REAL represents a real RIR convolved with clean audio. As such it is possible there is still a performance gap with actual ambisonics audio. I would suggest computing the direction-of-arrival error against STARSS'23 [1] which is fully recorded audio
* **Q6** [Minor] It would be interesting to ablate Nat-HEAR and the localisation tasks depending on the SNR [5-10, 10-20, 20-40]. This would establish how robust is GRAM to distractor noise.
* **Q6**: Some models seem to be missing from Fig 2.A (including Wav2Vec2 HuBERT and WavLM). Additionally, the caption of Fig 2 does not match its contents (the mentions two subfigures that are not there, and describes A in C, and B in A).


------
### Nitpicks (do not affect score, no need to follow up on these during rebuttal)

* **N1**:  In lines 253-256, could you list the number of parameters of each model
* **N2**: The points in Figs 2.A, 4, and 5 are pretty thick, can you make them smaller so that it's clear what is the actual performance of each setting?
* **N3**: Line 407, please indicate Table 5 is in the appendix.
* **N4**: At the end of section 4.1, please explicitly note in the main text that the results for the sampling factor and masking ratio are in Appendix F.
-----

[1] Shimada et al (2023) "STARSS23: An Audio-Visual Dataset of Spatial Recordings of Real Scenes with Spatiotemporal Annotations of Sound Events" NeurIPS Track on Datasets and Benchmarks.

---

> ### Author Response · Authors · 2025-11-24
> **Rebuttal 1/3**
>
> We thank the reviewer for taking the time to read and comment on our ICLR submission, as well as for the helpful suggestions to improve our work further. Point-by-point responses to the reviewer’s comments and questions are provided below. We thank you for taking our rebuttal into consideration.
>
> **W1:**  We thank the reviewer for suggesting to we also include downstream tasks focused on acoustics. In agreement with the reviewer’s suggestion, we **added a T60 Estimation task** to the localization tasks with the SC-5 and ESC-50 datasets in NatHEAR.We estimated T60 from the source RIRs and modelled this as a regression task. We compare the T60-estimation accuracy of GRAM-Binaural and GRAM-Ambisonics to GRAM-Clean to quantify the impact of naturalistic training on this task. Further, we compare performance to Spatial-AST.
>
> In our updated manuscript, **Figure 3B** shows the results of the T60 estimation tasks. We found that GRAM-Ambisonics and GRAM-Binaural estimate T60 significantly more accurately than Spatial-AST and GRAM-Clean on both ESC-50 and SC datasets (one-way ANOVA with post-hoc Tukey tests). These findings confirm that self-supervised models trained on naturalistic spatial sound scenes, such as GRAM learn holistic sound scene representations that also capture acoustic properties accurately. Even though Spatial-AST was trained with naturalistic spatial scenes as well, the supervised training pipeline targeted only sound localization and sound classification tasks, resulting in poor representations of T60.
>
>
> Regarding the reviewer’s suggestion to include a source separation task, we acknowledge that this would be a valuable addition. However, implementing this task would require a generative pipeline to reconstruct the separated streams, which falls outside the scope of the current work. We therefore plan to extend GRAM to include source separation tasks in future work.
>
> **Q1:** We apologize for the confusion. After the mel spectrogram conversion, the spectrograms have dimensions of 1001×128 (with one additional frame due to centering and padding). We subsequently perform zero-padding such that all spectrograms have dimension 1024×128 to ensure divisibility by 16 while maintaining an optimal patch count (64 patches rather than 63 with a dimension of 1008). We clarified this preprocessing step in the revised manuscript.
>
> **Q2:** We thank the reviewer for the suggestion to investigate the effect of in-batch sampling on convergence during model training. In our approach, we randomly selected 16 two-second segments from the full 10-second spectrogram after applying the naturalistic scene augmentations. While the probability of selecting exactly overlapping clips within a spectrogram is extremely low, we acknowledge that overlapping segments may introduce some gradient noise. To evaluate the potential performance impact of in-batch sampling, we trained a GRAM-Binaural model with an effective batch size of 512 (batch size of 32 with gradient accumulation over 16 steps) and compared downstream task performance on a subset of the HEAR and Nat-HEAR benchmark to a GRAM-Binaural model trained using a batch size of 32 with 16 samples per clip.
>
> In our updated manuscript **Figure 7** shows that there was no difference in performance or convergence between GRAM trained with in-batch sampling and GRAM trained without in-batch sampling.
>
>
> **Q3:** We thank the reviewer for pointing out the need for further clarification regarding GRAM-Clean and apologize for any confusion. Indeed, GRAM-Clean is a variant of GRAM trained on single-channel mel spectrograms, which serves as a baseline to quantify the benefits of incorporating naturalistic, spatial audio in the pre-training pipeline. We have revised the manuscript to clarify the training and purpose of GRAM-Clean.

---

> ### Author Response · Authors · 2025-11-24
> **Rebuttal 2/3**
>
> **Q4:** We thank the reviewer for emphasizing the need for clarification of the performance metric s(m). The s(m) metric was introduced as a summary statistic of model performance in the context of the SUPERB challenge [1]  to compare models on downstream tasks. This metric effectively ranks models based on their improvement over the baseline (here, HEAR-Naïve) and state-of-the-art performance, while accounting for task difficulty. That is, improvements on difficult tasks (tasks for which state-of-the-art performance was low) contribute more to the s(m) score than improvements on easy tasks (tasks for which state-of-the-art performance was already high) due to smaller denominators in the scoring function.
>
> For additional clarity, we have now also included average scores in the Results section. Below, Table 1 presents the average performance on the HEAR and NatHEAR benchmarks, respectively, for the reference of the reviewer. These average scores confirm that GRAMs learn robust audio representations with strong performance across downstream tasks.
>
>
> Table 1: Nat-HEAR and HEAR Benchmark scores
> |Model| Avg. Nat-HEAR  | Avg. HEAR
> |--|--|--|
> | GRAM-Binaural | 73.9  | 82.5
> | GRAM-Ambisonics  | 71.8 | 81.1
> | GRAM-Clean | 71.1  | 83.1
> | MAE | 54.2 |  65.5
> | SSAST | 44.9 |  56.2
> | MWMAE | 67.3  | 80.8
> | SSAM | 68.9 | 79.6
> | BEATs | 62.7 | 76.6
> | Wav2Vec2 | 46.2 |  55.7
> | HuBERT | 54.8 | 68.7
> | WavLM | 54.8 | 59.1
>
>
>  **Q5:**
>
>
> We thank the reviewer for pointing out the need to test the transferability of the representations learned by GRAM to recordings of real-world audio. In agreement with the suggestion of the reviewer, we therefore conducted experiments on the STARSS’23 dataset using the GRAM-Ambisonics model.
>
> As the STARSS’23 dataset includes multi-source sound scenes as well as moving sound sources and additionally has a different temporal resolution (labels per 100 ms versus GRAM’s output per 80 ms), we assessed the transferability of GRAM-Ambisonics using both the linear-head approach as used for the Nat-HEAR benchmark tasks and a more extensive full-model fine-tuning procedure to accommodate the differences in data.  Following the STARSS [4] convention, we reported at ER20◦,F20◦, LR_CD and LE_CD. Two metrics are referred to as location-aware detection and are error rate
> (ER20◦ ) and F-score (F20◦ ) in one-sec non-overlapping segments. LE_CD expresses the average angular
> difference between the same class’s predictions and references. LR_CD tells the true positive rate of
> how many of these localization estimates were detected in a class out of the total number of class
> instances.
>
> As expected based on the different nature of the STARSS’23 data, the direct transfer of GRAM-Ambisonics representations using a linear-head approach results in poor performance (localization error of 47.1 °, Table 3).  However, it should be noted that the poor performance was mainly caused by a failure to detect the event classes ‘door’ and ‘bell’ (F-score 0.0, Table 4), which automatically results in a localization error of 180°. Removing these classes reveals an average localization error of  22.9°. This suggests that the model's representations are effective for a majority of the classes and our pre-trained representations transfer accurately to challenging real-life recordings.
>
> To accommodate the different nature of the STARSS’23 dataset, we also investigated full model fine-tuning on STARSS’23 to allow the model to adapt to its unique challenges. As shown in Table 3, fine-tuning yields a dramatic improvement, bringing the LE_CD down to 18.6° and closing the performance gap considerably
>
> Furthermore, the fine-tuning dynamics illustrated in **Figure 4** in our updated manuscript demonstrate that our pre-trained representations enable rapid and effective adaptation to the real world. The fine-tuned model achieves high performance within just 20 epochs, significantly outperforming a randomly initialized model trained from scratch, which plateaus at a lower performance level regardless of hyperparameter tuning. This demonstrates that our pre-training provides a robust and transferable initialization for complex, real-world tasks. We will include the STARSS’23 evaluation and results in the camera-ready manuscript.

---

> ### Author Response · Authors · 2025-11-24
> **Rebuttal 3/3**
>
> Table 3: STARSS23 Results, State-of-the-art (SOTA) model is taken from [2], Baseline is taken from [3].
>
> | Model                    | ER₂₀° ↓ | F₂₀° ↑ | LE_CD ↓ | LR_CD ↑ |
> | ------------------------ | ------- | ------ | ------- | ------- |
> | **Native Sampling Rate** |         |        |         |         |
> | SOTA                     | 0.42    | 59.0%  | 13°     | 72.0%   |
> | Baseline                 | 0.57    | 29.9%  | 22°     | 47.7%   |
> | **Upsampled**            |         |        |         |         |
> | GRAM-Amb. (Full model fine-tuning)    | 0.51    | 41.4%  | 18.6°   | 60.5%   |
> | GRAM-Amb. (Linear head)      | 0.59    | 23.8%  | 47.1°   | 35.0%   |
> | Baseline (Reprod.)       | 0.62    | 28.3%  | 23.7°   | 45.7%   |
>
> Table 4: Class-wise metrics for GRAM-Ambisonics Linear head
> | Class | ER₂₀° ↓ | F₂₀° ↑ | LE_CD ↓ | LR_CD ↑ |
> |-------|-------------------|--------------------|-------------------------|--------------------|
> | 0     | 0.59 [0.56, 0.64] | 0.52 [0.40, 0.64]  | 25.27 [15.50, 34.99]    | 0.82 [0.78, 0.86]  |
> | 1     | 0.59 [0.56, 0.64] | 0.63 [0.56, 0.70]  | 15.17 [13.77, 16.61]    | 0.74 [0.70, 0.78]  |
> | 2     | 0.59 [0.56, 0.64] | 0.20 [0.03, 0.37]  | 18.19 [13.20, 23.17]    | 0.19 [0.09, 0.28]  |
> | 3     | 0.59 [0.56, 0.64] | 0.01 [-0.01, 0.03] | 36.21 [27.53, 48.90]    | 0.37 [0.06, 0.71]  |
> | 4     | 0.59 [0.56, 0.64] | 0.36 [0.28, 0.44]  | 17.63 [13.83, 21.18]    | 0.40 [0.31, 0.48]  |
> | 5     | 0.59 [0.56, 0.64] | 0.30 [0.16, 0.45]  | 20.54 [16.75, 24.11]    | 0.43 [0.23, 0.65]  |
> | 6     | 0.59 [0.56, 0.64] | 0.01 [-0.00, 0.02] | 37.45 [27.60, 48.61]    | 0.05 [0.02, 0.10]  |
> | 7     | 0.59 [0.56, 0.64] | 0.00 [0.00, 0.00]  | 180.00 [180.00, 180.00] | 0.00 [0.00, 0.00]  |
> | 8     | 0.59 [0.56, 0.64] | 0.24 [0.11, 0.39]  | 37.12 [22.14, 51.88]    | 0.63 [0.51, 0.73]  |
> | 9     | 0.59 [0.56, 0.64] | 0.40 [0.31, 0.50]  | 20.91 [18.99, 22.74]    | 0.66 [0.56, 0.78]  |
> | 10    | 0.59 [0.56, 0.64] | 0.28 [0.04, 0.50]  | 13.19 [8.69, 20.37]     | 0.19 [-0.02, 0.38] |
> | 11    | 0.59 [0.56, 0.64] | 0.00 [0.00, 0.00]  | 180.00 [180.00, 180.00] | 0.00 [0.00, 0.00]  |
> | 12    | 0.59 [0.56, 0.64] | 0.13 [-0.14, 0.43] | 10.31 [-490.76, 176.36] | 0.07 [-0.08, 0.22] |
>
>
>
> **Q6**:  We thank the reviewer for this suggestion to evaluate the robustness of GRAM to varying SNR. To investigate the localization performance of our models under the distractor noise, we conducted an ablation study on the NatHEAR benchmark tasks using various noise levels: high(5-10 dB), medium (10-20 dB), and low (20-40 dB) SNR.
>
> The results, illustrated in **Figure 8** of our updated manuscript, reveal that GRAM-Ambisonics is remarkably invariant to low SNR, while both GRAM-Binaural and Spatial-AST exhibited substantial performance degradation at low SNR. These findings show that the representations learned by GRAM-Ambisonics are highly robust to low SNR conditions for localization.
>
> Unfortunately, we were unable to perform the experiment for the full Nat-HEAR dataset due to significant storage overhead. However, we plan on adding this experiment for our camera-ready version after acquiring an expanded storage facility.
>
>
>
> **Q7:** We thank the reviewer for pointing out the need to also include the other models in Figure 2. And we apologize for any confusion. We now include all models trained on AudioSet, but exclude models trained on different data, as this would not be a fair comparison. These are Wav2Vec2, HuBERT, and WavLM trained on LibriSpeech and LibriLight. Furthermore, we updated the caption.
>
>
> [1] Feng, Tzu-hsun, et al. "Superb@ slt 2022: Challenge on generalization and efficiency of self-supervised speech representation learning." 2022 IEEE Spoken Language Technology Workshop (SLT). IEEE, 2023.
>
> [2] Qing Wang, Ya Jiang, Shi Cheng, Maocheng Hu, Zhaoxu Nian, Pengfei Hu, Zeyan Liu, Yuxuan
> Dong, Mingqi Cai, Jun Du, and Chin-Hui Lee. The nerc-slip system for sound event localization
> and detection of dcase2023 challenge. Technical report, DCASE2023 Challenge, June 2023
>
> [3] Kazuki Shimada, Archontis Politis, Parthasaarathy Sudarsanam, Daniel Krause, Kengo Uchida,
> Sharath Adavanne, Aapo Hakala, Yuichiro Koyama, Naoya Takahashi, Shusuke Takahashi, Tuomas
>
> [4] Shimada et al (2023) "STARSS23: An Audio-Visual Dataset of Spatial Recordings of Real Scenes with Spatiotemporal Annotations of Sound Events" NeurIPS Track on Datasets and Benchmarks.

---

> > ### Comment · Reviewer_zoEh · 2025-11-26
> >
> > I would like to thank the authors for their detailed and thoughtful rebuttal.
> >
> > In order to streamline the discussion, I will only follow up on topics where I have further questions or nitpicks. (That is if I don't follow up on a topic, it means I am satisfied with the authors' response).
> >
> > **W1**
> >
> > > Regarding the reviewer’s suggestion to include a source separation task, we acknowledge that this would be a valuable addition. However, implementing this task would require a generative pipeline to reconstruct the separated streams, which falls outside the scope of the current work.
> >
> > I suggested T60 estimation _or_ source separation. Implementing both would be too much to demonstrate generalisability to acoustic downstream tasks.
> >
> > **T60 Estimation on SC-5 and ESC-50**
> >
> > Figure 2b is missing units on the y-axis, I assume it refers to seconds. That means GRAM achieves ~20ms median absolute error on SC-5 and ~40ms, correct? (Incidentally, I would suggest adding a table to the appendix with the actual medians, so other practitioners can easily compare their results in the future).
> >
> > Additionally, what model represents the right-most box-plot? GRAM-Clean? (To my eye they seem to have different shades of grey).
> >
> > What are the T60 ranges of the rooms in SC-5 and ESC-50?
> >
> > Since T60 is frequency dependent, are you reporting an aggregate across many frequencies or simply estimating T60 for a particular frequency?
> >
> > **Phase Encoding**
> >
> > Reviewer rz9E brings up an interesting point that the phase information is currently discarded in GRAM. Clearly, this does not seem to affect the downstream tasks, where GRAM exhibits good performance (including, perhaps surprisingly, T60 estimation).
> >
> > Still, I wonder if you could run a quick experiment including the reactive components of the intensity vectors to the inputs of GRAM-Ambisonics to check whether there is any improvement/degradation in the downstream tasks when including phase information.
> >
> > Given the limited time for this, perhaps some combination of direction-of-arrival, T60 and a few HEAR, NAT-HEAR tasks would suffice.
> >
> > **Ablations on in-batch sampling, SNR, and experiments with STARSS'23**
> >
> > Thank you for running these ablations and additional experiments in such a short time. I believe these strengthen the paper.

---

> > > ### Author Response · Authors · 2025-12-01
> > >
> > > We'd like to thank the reviewer for the follow-up questions on the $T_{60}$ estimation tasks.
> > >
> > > **$T_{60}$ Estimation on SC-5 and ESC-50** We have now added the correct units on the y-axis in Figure 2b, and included a section with our results in Appendix D. The rightmost plot indeed represents GRAM-Clean. We have also corrected the legend color. We apologize for the mistake.
> > >
> > > **$T_{60}$ Ranges of the Rooms in SC-5 and ESC-50:** The $T_{60}$ values range from [0.05, 0.7] seconds, with the mode of the distribution around $\sim0.2$ seconds. We have added histogram plots for the estimated $T_{60}$ values for both the SC-5 and ESC-50 datasets in Appendix D, Figure 6.
> > >
> > > **$T_{60}$ Reporting:** Because the source room impulse responses were available, we used Schroeder's method [1] for $T_{60}$ estimation. This method estimates the decay curve of the acoustic energy over time rather than estimating $T_{60}$ across or for a particular frequency band. We then utilized the computed decay curve to extrapolate to $-60$ dB. Specifically, we used the Pyroomacoustics [2] package to perform the estimation.
> > >
> > > **Reactive Components:** We are currently training the GRAM-Ambisonics model with mel spectrograms + active + reactive intensity vectors for $50,000$ steps. We plan to compare these results with the standard GRAM-Ambisonics model (also $50,000$ steps) on the CD, ESC-50, LC, and VL subtasks for the HEAR and Nat-HEAR suites. Furthermore, we are going to run all the localization and $T_{60}$ estimation tasks. We will report the results as soon as the training run is finalized.
> > >
> > > [1] M. R. Schroeder. New method of measuring reverberation time. The Journal of the Acoustical Society
> > > of America, 37(6 Supplement):1187–1188, 06 1965. ISSN 0001-4966. doi: 10.1121/1.1939454.
> > > URL https://doi.org/10.1121/1.1939454
> > >
> > > [2] Robin Scheibler, Eric Bezzam, and Ivan Dokmani ´c. Pyroomacoustics: A Python package for audio room simulation and array processing algorithms. In 2018 IEEE International Conference on Acoustics, Speech and Signal Processing (ICASSP), pp. 351–355, 2018. doi: 10.1109/ICASSP. 2018.8461310.

---

> > > > ### Author Response · Authors · 2025-12-02
> > > >
> > > > Herewith the downstream performance of GRAM-Ambisonics with reactive components added. Table 1 and Table 2 demonstrate that GRAM-Ambisonics with active intensity vectors perform better on ESC-50 and the LC subtask on HEAR and Nat-HEAR. Furthermore, we did not find a significant difference between RT60 estimation on the SC subtask and the ESC-50 subtask (Table 3). Lastly, the localization performance of both models is on par (Table 4).
> > > >
> > > > These results suggest that adding reactive components does not improve RT60 estimation and localization performance, and possibly hinders the performance on HEAR and Nat-HEAR downstream tasks. This finding also suggests that GRAM-Ambisonics learn the required localization and reverberation cues from mel spectrograms and active intensity vectors, which is in line with current literature that utilizes similar input features for ambisonics [1,2]
> > > >
> > > >
> > > > Table 1: HEAR Results
> > > > |Model|ESC-50|LC|CD|VL
> > > > |---|---|---|---|---|
> > > > GRAM-Ambisonics Mel + Active|**82.3** +- 1.9 |**78.2** +- 1.0 |**70.3** +- 0.9|**43.8** +- 2.7
> > > > |GRAM-Ambisonics Mel + Active + Reactive|78.5 +- 2.5 |77.0 +- 1.6|70.0 +- 1.1| 41.8 +- 2.9
> > > >
> > > > Table 2: NatHEAR Results
> > > > |Model|ESC-50|LC|CD|VL
> > > > |---|---|---|---|---|
> > > > |GRAM-Ambisonics Mel + Active|**74.5** +- 2.2|**71.9** +- 1.2|**56.4** +- 0.8 |**26.8** +- 2.7
> > > > |GRAM-Ambisonics Mel + Active + Reactive|72.0 +- 1.6|67.4 +- 1.8|55.2 +- 0.9 |25.9 +- 2.6
> > > >
> > > > Table 3: Localization Results, (Median DOE)
> > > > |Model|ESC-50|SC
> > > > |---|---|---|
> > > > |GRAM-Ambisonics Mel + Active|**23.4**|**15.2**
> > > > |GRAM-Ambisonics Mel + Active + Reactive|23.9|17.8
> > > >
> > > > Table 4: RT60 Estimation Results (Mean Absolute Error)
> > > > |Model|ESC-50|SC
> > > > |---|---|---|
> > > > |GRAM-Ambisonics Mel + Active|**0.049** |0.025
> > > > |GRAM-Ambisonics Mel + Active + Reactive|0.057|0.025
> > > >
> > > > [1] Shimada et al (2023) "STARSS23: An Audio-Visual Dataset of Spatial Recordings of Real Scenes with Spatiotemporal Annotations of Sound Events" NeurIPS Track on Datasets and Benchmarks.
> > > >
> > > > [2] Nguyen, T. N. T., Watcharasupat, K. N., Nguyen, N. K., Jones, D. L., & Gan, W.-S. (2022). SALSA: Spatial Cue-Augmented Log-Spectrogram Features for Polyphonic Sound Event Localization and Detection. IEEE/ACM Transactions on Audio, Speech, and Language Processing, 30, 1749–1762. https://doi.org/10.1109/TASLP.2022.3173054

---

### Author Response · Authors · 2025-11-25
**Reflecting on changes**

We sincerely thank all anonymous reviewers for their valuable time and thoughtful feedback on our ICLR submission. We have carefully considered each comment and made substantial revisions to our manuscript. In this global rebuttal, we would like to highlight the key updates we have made.

## Evaluation on STARSS’23 Dataset
---
We have incorporated STARSS23 [1] into our evaluation setup to assess the transferability of naturalistic training to challenging real-world sound scenes with polyphonies, moving sources, and interference noise. The results are shown in **Table 3** and **Figure 4**. We find that minimally fine-tuning GRAM on the STARSS’23 dataset results in strong localization performance (localization error of 18.6 degrees), highlighting the transferability of GRAM’s spatial audio representations to real-world recordings.


## Extension of Nat-HEAR Benchmark with RT60 Estimation
---
Following reviewer suggestions, we have extended our Nat-HEAR benchmark to include an **RT60 estimation task**.
Figure 3B in our updated manuscript demonstrates that GRAM-Ambisonics and GRAM-Binaural estimate RT60 significantly more accurately than both Spatial-AST and GRAM-Clean on the ESC-50 and SC datasets (one-way ANOVA with post-hoc Tukey tests). These findings confirm that self-supervised models trained on naturalistic spatial sound scenes, such as GRAM, learn holistic sound scene representations that accurately capture acoustic properties.

## Extended Ablation Studies
---
We have updated our ablation studies to further investigate the effect of added noise on Nat-HEAR localization and classification performance. We have completed the experiments for Nat-HEAR localization and added the results in **Figure 8**. We plan to include the results on Nat-HEAR classification in our camera-ready version.

The results reveal that GRAM-Ambisonics is remarkably invariant to low SNR conditions, while both GRAM-Binaural and Spatial-AST exhibit significant performance degradation at low SNR. These findings demonstrate that the representations learned by GRAM-Ambisonics are highly robust to low SNR conditions.

### In-Batch Sampling Analysis
----
We have added **Figure 7**, which illustrates the potential performance impact of in-batch sampling. We trained a GRAM-Binaural model with an effective batch size of 512 (batch size of 32 with gradient accumulation over 16 steps) and compared downstream task performance on a subset of the HEAR and Nat-HEAR benchmarks to a GRAM-Binaural model trained using batch size 32 with 16 samples per clip.

We did not find a significant difference in performance or convergence between GRAM trained with in-batch sampling and GRAM trained without in-batch sampling.

## Comparison with Supervised Baselines
-----

To provide additional context for the performance of our self-supervised GRAM models, we extended the HEAR and Nat-HEAR benchmarks with two supervised baselines: Spatial-AST [2] (trained on spatialized AudioSet with simulated RIRs) and PASST [3] (trained on AudioSet). The results are presented in Table 1 and Table 7 of our updated manuscript.

When compared directly to the most relevant baseline, Spatial-AST, our GRAM models demonstrate several key advantages:
* Data Efficiency: GRAMs achieve higher performance with less data, as evidenced by the learning curves in **Figure 2A**.
* Localization Capabilities: As shown in **Figure 3A**, GRAMs learn robust spatial representations.
* Stronger Performance: GRAMs achieve higher aggregate scores on both the HEAR and Nat-HEAR benchmarks (Tables 1 & 7).
* Robustness: GRAMs are more resilient to challenging acoustic conditions such as noise and reverberation (**Figure 2B**).

Critically, GRAM-Binaural learns these powerful localization cues without explicit phase-based input or localization supervision, demonstrating its ability to discover useful spatial representations from the mel spectrogram alone.

We believe these revisions substantially address the reviewers' concerns and strengthen our manuscript. We remain open to any additional feedback and are grateful for the opportunity to improve our work.

[1] Shimada, K., Politis, A., Sudarsanam, P., Krause, D., Uchida, K., Adavanne, S., Hakala, A., Koyama, Y., Takahashi, N., Takahashi, S., Virtanen, T., & Mitsufuji, Y. (2023). STARSS23: An Audio-Visual Dataset of Spatial Recordings of Real Scenes with Spatiotemporal Annotations of Sound Events (No. arXiv:2306.09126). arXiv. https://doi.org/10.48550/arXiv.2306.09126

[2] Zheng, Z., Peng, P., Ma, Z., Chen, X., Choi, E., & Harwath, D. (2025). BAT: Learning to Reason about Spatial Sounds with Large Language Models (No. arXiv:2402.01591). arXiv. https://doi.org/10.48550/arXiv.2402.01591

[3] Koutini, K., Schlüter, J., Eghbal-zadeh, H., & Widmer, G. (2021, October 11). Efficient Training of Audio Transformers with Patchout. arXiv.Org. https://doi.org/10.21437/Interspeech.2022-227

---

### Meta-Review · Area_Chair_naYK · 2026-01-05

**Summary:**

The paper proposes GRAM, a multi-channel masked auto-encoder aimed at learning general-purpose spatial audio representations. The authors introduce two variants (Binaural and Ambisonics) trained on simulated data and evaluate them on an extended benchmark suite (Nat-HEAR). After a thorough review process and rebuttal phase, I am recommending a rejection for this submission. While the empirical performance on the specific tested formats is competitive, there are fundamental disagreements regarding the scope and generalizability of the model that were not resolved, especially the concerns from Reviewer rz9E.

**Reviewer Concerns:**

The authors provided a detailed rebuttal that addressed several concerns, yet core issues remain unsolved:

Addressed Concerns:

+ Real-world validation: In response to requests from zoEh and ESjb for real-world data validation, the authors conducted experiments on the STARSS'23 dataset. This demonstrated that the model could transfer to real recordings, partially alleviating concerns about the reliance on simulated training data.
+ Downstream Tasks: The authors added a T60 estimation task to the Nat-HEAR benchmark to address zoEh's concern about the lack of acoustic property evaluation. zoEh explicitly stated they were satisfied with this addition and the subsequent ablation studies regarding SNR and in-batch sampling.

Outstanding Concerns:
- Generalization across Geometries: Reviewer rz9E maintained a strong objection regarding the fixed input formats. The reviewer correctly points out that broad data in the context of spatial audio implies diversity in hardware and microphone array geometry (linear, circular, ad-hoc arrays). The authors argued that foundation model refers to the breadth of downstream tasks, but I agree with rz9E that a model requiring retraining for every new microphone configuration falls short of being a spatial foundation model. This is a significant limitation for real-world deployment outside of standard Ambisonics/Binaural contexts.
- Input Representation: Reviewer rz9E remained unconvinced by the use of spectrograms. The reviewer noted that for spatial tasks, raw waveforms are often preferred to preserve fine-grained phase cues essential for localization, which spectrograms discard or obscure.

**Reviewer Scores:**

Based on the above issues, I think all the reviewers will maintain their scores.

---

### Decision · Program_Chairs · 2026-01-26

Reject